# Inhibition of myeloid-derived suppressor cell arginase-1 production enhances T-cell-based immunotherapy against *Cryptococcus neoformans* infection

Ya-Nan Li[1,2,4], Zhong-Wei Wang[2,4], Fan Li[2,4], Ling-Hong Zhou[3,4], Yan-Shan Jiang[1,2], Yao Yu[2], Hui-Hui Ma[2], Li-Ping Zhu[3], Jie-Ming Qu [1✉] & Xin-Ming Jia [2✉]

Cryptococcosis is a potentially lethal disease that is primarily caused by the fungus *Cryptococcus neoformans*, treatment options for cryptococcosis are limited. Here, we show glucuronoxylomannan, the major polysaccharide component of *C. neoformans*, induces the recruitment of neutrophilic myeloid-derived suppressor cells in mice and patients with cryptococcosis. Depletion of neutrophilic myeloid-derived suppressor cells enhances host defense against *C. neoformans* infection. We identify C-type lectin receptor-2d recognizes glucuronoxylomannan to potentiate the immunosuppressive activity of neutrophilic myeloid-derived suppressor cells by initiating p38-mediated production of the enzyme arginase-1, which inhibits T-cell mediated antifungal responses. Notably, pharmacological inhibition of arginase-1 expression by a specific inhibitor of p38, SB202190, or an orally available receptor tyrosine kinase inhibitor, vandetanib, significantly enhances T-cell mediated antifungal responses against cryptococcosis. These data reveal a crucial suppressive role of neutrophilic myeloid-derived suppressor cells during cryptococcosis and highlight a promising immunotherapeutic application by inhibiting arginase-1 production to combat infectious diseases.

[1] Department of Pulmonary and Critical Care Medicine, Ruijin Hospital, Shanghai Jiao Tong University School of Medicine, Shanghai 200025, China. [2] Clinical Medicine Scientific and Technical Innovation Center, Shanghai Tenth People's Hospital, Tongji University School of Medicine, Shanghai 200092, China. [3] Department of Infectious Diseases, Shanghai Key Laboratory of Infectious Diseases and Biosafety Emergency Response, National Medical Center for Infectious Diseases, Huashan Hospital, Fudan University, Shanghai 200025, China. [4]These authors contributed equally: Ya-Nan Li, Zhong-Wei Wang, Fan Li, Ling-Hong Zhou. ✉email: jmqu0906@163.com; jiaxm@tongji.edu.cn

Cryptococcosis is primarily caused by *Cryptococcus neoformans*, and it is life-threatening in immunocompromised hosts. Initial pulmonary infection with *C. neoformans* typically spreads systemically to the central nervous system, which presents as meningitis. An estimated 223,100 cases of cryptococcal meningitis occur globally annually, which leads to about 181,100 deaths[1]. Clinical and experimental data have established that CD4+T-cell-mediated immunity (CMI) is essential for the control of cryptococcal infection since this disease occurs mainly in those with impaired CMI including HIV-infected individuals, with an incidence of 30% and a mortality of 30 to 60%[2]. Notably, anti-retroviral therapy (ART), which is applied to restore CD4+T-cell numbers in AIDS patients, can dramatically reduce the mortality and morbidity of AIDS-associated cryptococcal meningitis[3]. However, a hallmark of infection with *C. neoformans* is the depression of the immune system characterized by poor CMI and pro-inflammatory responses. Therefore, there is an urgent need to develop new immunotherapeutic strategies for enhancing the function of CD4+ T-cells to combat cryptococcosis.

Fungal cell wall glycans and exopolysaccharides are the first point of physical contact in pathogen–host interactions. Pattern recognition receptors (PRRs), primarily C-type lectin receptors (CLRs), on innate immune cells recognize these glycans and eliminate fungi[4–6]. However, *Cryptococcus* cells prevent recognition and phagocytosis by masking themself in an exopolysaccharide-based capsule composed primarily of glucuronoxylomannan (GXM)[7]. Numerous lines of studies indicate that GXM exhibits potent immunosuppressive properties of inhibiting T-cell responses[8,9]. CMI by CD4+T$_H$1 cells is the primary host defense response against cryptococcosis, which is evidenced clinically by the high susceptibility of individuals with reduced CMI to *C. neoformans* infections[10]. The drastic decrease in CMI responses to *C. neoformans* infection is presumably due to the capacity of GXM to induce T-cell apoptosis by upregulating Fas-L expression in macrophages[11,12] or impair T-cell activation[8,11,13–15], likely via the downregulating of major histocompatibility complex (MHC) class II and B7 expression in macrophages[16,17].

The importance of a heterogeneous group of immature myeloid cells, termed myeloid-derived suppressor cells (MDSCs), that functionally suppress T-cell responses[18] is highlighted in infectious diseases, including infection with *Mycobacterium tuberculosis*[19–21], hepatitis C virus[22,23] and parasites[24]. MDSCs are further classified into monocytic (M, Ly6C$^{hi}$Ly6G$^-$) or polymorphonuclear (PMN, Ly6C$^{low}$Ly6G$^+$) subsets. MDSCs produce various immunosuppressive mediators, including arginase-1 (ARG1) and nitric oxide (NO), to disassemble L-arginine, which is required for T-cell proliferation[25]. MDSCs produce reactive oxygen species (ROS) to disrupt T-cell function[26]. We showed that the release of GXM by *C. neoformans* increased the recruitment of neutrophilic MDSCs thereby aggravating this infectious disease. CLEC2D (also known as Lectin-like transcript 1, LLT1), a CLR, recognized GXM and promoted neutrophilic MDSCs via p38-mediated production of ARG1, which impaired T-cell responses against *C. neoformans* infection. We also validated that pharmacological inhibition of MDSC-derived ARG1 production significantly enhanced T-cell-mediated antifungal responses against *C. neoformans* infection. Together, our data indicate that neutrophilic MDSCs play a role in respect of amplification of cryptococcal lung disease and inhibiting MDSC-derived ARG1 production is a promising immunotherapeutic strategy for treating this disease.

## Results

**C. neoformans infection induces the recruitment of MDSCs, especially PMN-MDSCs, to exert deleterious effects**. To evaluate myeloid cell responses during *C. neoformans* infection, lung-infiltrating CD11b+ cells were analyzed in mice after a clinically relevant route of infection with *C. neoformans* serotype A strain H99, which is the most clinically prevalent in immunocompromised individuals. A significant increase was observed in the lung CD11b+ population expressing the granulocytic marker GR1 (Fig. 1a and Supplementary Fig. 1a, b), which defines MDSCs, accompanied by the worsening of pulmonary *C. neoformans* infection (Supplementary Fig. 1c, d). Further analysis showed that pulmonary infection with *C. neoformans* time-dependently triggered the recruitment of PMN-MDSCs, and M-MDSCs were not induced in mice (Fig. 1b and Supplementary Fig. 1b). PMN-MDSCs may be distinguished from neutrophils by surface markers, including CD244 and CD115[27]. We found that *C. neoformans* infection significantly triggered the recruitment of CD244+ PMN-MDSCs or CD115+ PMN-MDSCs (Fig. 1c). We further quantified human MDSCs expressing the myeloid markers CD11b+, CD33+, and CD15+ as described previously[25] to determine the clinical relevance of the murine studies (Supplementary Fig. 1e). We observed that non-HIV 15 patients with pulmonary cryptococcosis and 14 patients with cryptococcal meningitis (Supplementary Table 1) significantly accumulated PMN-MDSCs, but not M-MDSCs, in PBMCs compared to healthy controls (Fig. 1d and Supplementary Fig. 1f). Among these patients, seven with pulmonary cryptococcosis and nine with cryptococcal meningitis are immunosuppressed or immunocompromised. However, there was no significant difference about the accumulation of PMN-MDSCs between immunocompetent and immunosuppressed or immunocompromised patients (Supplementary Fig. 1g). These data indicated that *C. neoformans* infection induced the recruitment of neutrophilic MDSCs in mice and human subjects.

To determine the role of neutrophilic MDSCs during *C. neoformans* infection, we successfully depleted PMN-MDSCs, but not M-MDSCs, using an anti-Ly6G Ab (Fig. 1e and Supplementary Fig. 1h). Depletion of PMN-MDSCs in mice led to a higher survival rate and lower fungal burden than those who received control IgG after infection with *C. neoformans* (Fig. 1f, g). Histological analysis of the infected lungs on day 14 showed that PMN-MDSC depletion also led to a low number of yeasT-cells, which were mostly encapsulated in the granulomatous tissues (Fig. 1h). To confirm the specificity of anti-Ly6G-mediated PMN-MDSC depletion, anti-DR5 Ab was administered on the same schedule as the specific PMN-MDSC depletion antibody[28]. A significant decrease in the percentage of PMN-MDSCs, but not M-MDSCs, was observed in the lungs of mice after anti-DR5 treatment (Fig. 1i and Supplementary Fig. 1i). PMN-MDSC depletion led to a lower fungal burden, decreased inflammation, and fewer yeast-cells in the lungs of mice after infection with *C. neoformans* (Fig. 1j, k). These data suggested that pulmonary infection with *C. neoformans* induced the recruitment of MDSCs, especially PMN-MDSCs, to exert deleterious effects on host antifungal immunity.

**GXM induces the expansion and activation of MDSCs, especially PMN-MDSCs, to suppress T-cells and aggravate C. neoformans infection**. To determine whether capsular GXM was required for the induction of MDSCs during *C. neoformans* infection, we compared the influence of the *cap59Δ* mutant (Cap59) hypocapsular strain[29] to its parental encapsulated strain H99 on MDSC recruitment. We found that pulmonary infection with Cap59 significantly impaired the accumulation of MDSCs, especially PMN-MDSCs, compared to H99 infection (Fig. 2a and Supplementary Fig. 2a). We further found that pulmonary infection with the encapsulated strain H99 dramatically increased the production of ARG1 and NO, but not ROS, in the lungs (Fig. 2b). However, pulmonary infection with the hypocapsular strain Cap59 significantly impaired *Arg1* expression but had no influence on NO or ROS production in the lungs (Fig. 2b).

We extracted extracellular polysaccharides, which were primarily composed of GXM, from H99 cells as previously reported[30].

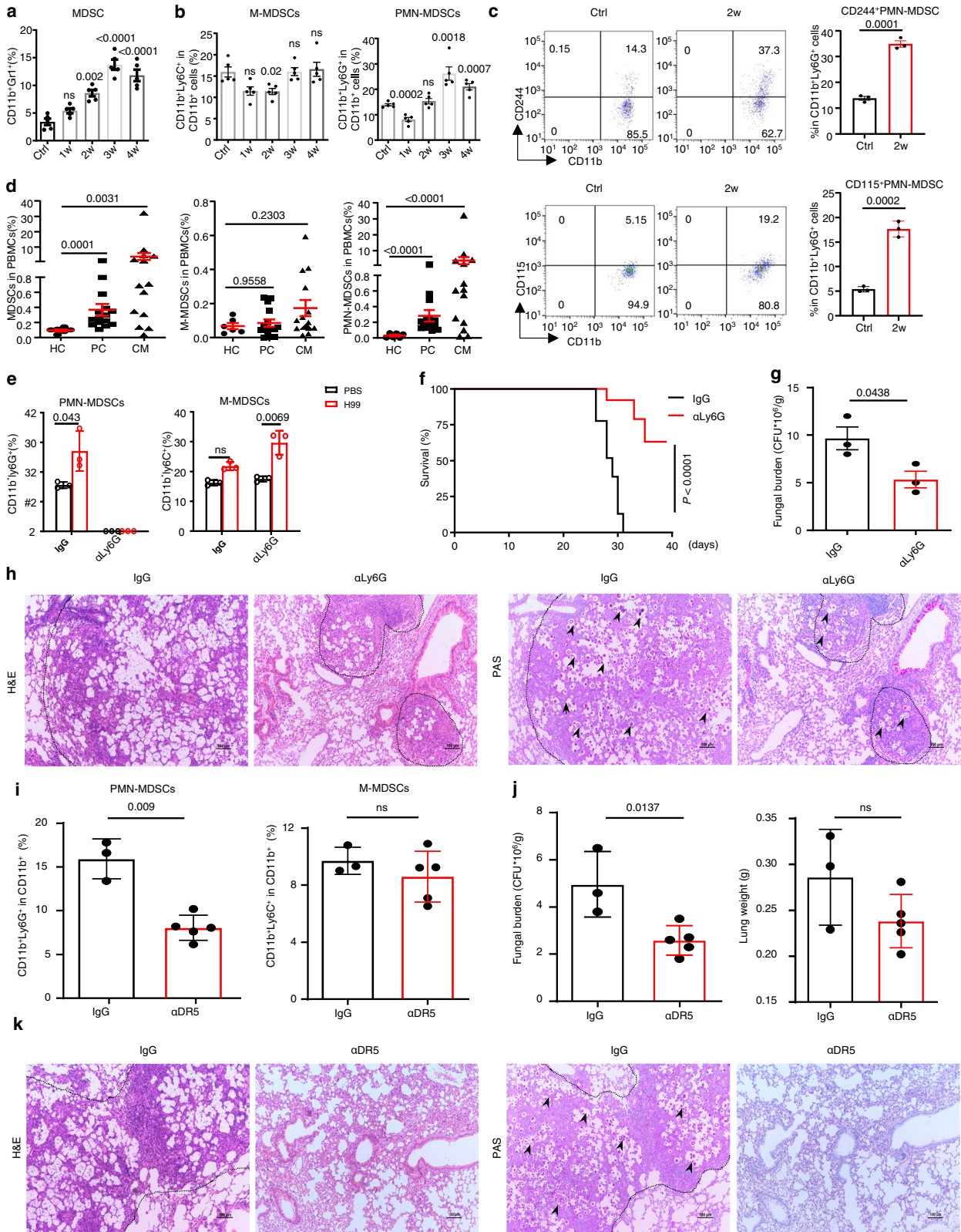

Notably, GXM treatment significantly increased the percentage of MDSCs, including PMN-MDSCs and M-MDSCs, in the lungs of mice (Fig. 2c and Supplementary Fig. 2b). GXM treatment prominently increased ARG1 and ROS production in murine lungs (Fig. 2d). However, GXM pretreatment prominently increased the percentage of MDSCs, especially PMN-MDSCs, in the lungs of Cap59-infected mice (Supplementary Fig. 2c, d).

Notably, GXM pretreatment significantly increased the fungal burden in the lungs of Cap59-infected mice (Supplementary Fig. 2d). These data suggested that GXM induced the recruitment of MDSCs, especially PMN-MDSCs, to aggravate *C. neoformans* infection.

We further evaluated the GXM-induced tolerogenic activities of MDSCs both ex vivo and in vivo (Fig. 2e). We found that

**Fig. 1 _C. neoformans_ infection induces the recruitment of MDSCs, especially PMN-MDSCs, to exert deleterious effects. a, b** The percentage of CD11b$^+$Gr1$^+$ MDSCs, CD11b$^+$Ly6G$^+$Ly6C$^{low}$ M-MDSCs, and CD11b$^+$Ly6G$^-$Ly6C$^{high}$ PMN-MDSCs in mice, which were intratracheally infected without (Ctrl) or with a single exposure of _C. neoformans_ strain H99 ($1 \times 10^3$ CFU/mouse) for the indicated week (w). **c** The percentage of CD11b$^+$Ly6G$^+$CD244$^+$ and CD11b$^+$Ly6G$^+$CD115$^+$ PMN-MDSCs in H99-infected mice on week 2 (2w) as described in **a, b. d** The percentage of HLA-DR$^-$CD11b$^+$CD33$^+$CD15$^+$CD14$^-$ PMN-MDSCs and HLA-DR$^-$CD11b$^+$CD33$^+$CD15$^-$CD14$^+$ M-MDSCs in PBMCs from healthy controls (HC, $n = 6$) and patients with pulmonary cryptococcosis (PC, $n = 15$) or cryptococcal meningitis (CM, $n = 14$). **e–h** The percentage of M-/PMN-MDSCs (**e**), survival curve (**f**), fungal burden (**g**), and representative histological images with hematoxylin-eosin (H&E) and Periodic Acid-Schiff (PAS) staining (**h**) of lungs from H99-infected mice, which were intraperitoneally treated with anti-Ly6G antibody (αLy6G) or control IgG twice a week. Scale bars = 100 μm. **i–k** The percentage of M-/PMN-MDSCs (**i**), fungal burden and lung weight (**j**), and representative histological images with H&E and PAS staining (**k**) of lungs from H99-infected mice, which were intraperitoneally treated with anti-DR5 antibody (αDR5) or control IgG twice a week. Scale bars = 100 μm. Dash lines indicate granulomatous lesions and arrowheads indicate yeasT-cells. Data were presented as mean ± SEM; $n = 3$ (**c, e, g, i** IgG group, **j** IgG group), $n = 5$ (**a, i** αDR5 group, **j** αDR5 group), or $n = 8$ (**f**) biologically independent samples. Data were analyzed by unpaired two-sided Student's _t_-test in **a–d, g, i, j**, one-way ANOVA adjusted for multiple comparisons in **e**, or two-sided log-rank (Mantel–Cox) tests in **f**. Source data are provided as a Source Data file.

MDSCs sorted from the lungs of GXM-treated mice strongly suppressed CD4$^+$ and CD8$^+$T-cell proliferation in a dose-dependent manner (Fig. 2f and Supplementary Fig. 2e). Moreover, we found that the adoptive transfer of MDSCs from the lungs of GXM-treated mice significantly increased the percentage of MDSCs, especially PMN-MDSCs, and fungal burdens in the lungs of Cap59-infected mice (Fig. 2g, h and Supplementary Fig. 2f). Notably, the adoptive transfer of MDSCs sorted from the lungs of GXM-treated mice significantly increased the expression of _Arg1_, but not _Nos2_ (the gene encoding inducible NO synthase), in the lungs of Cap59-infected mice (Fig. 2i). These data indicated that GXM induced the tolerogenic activity of MDSCs to inhibit T-cell-mediated antifungal responses and aggravate _C. neoformans_ infection.

We further evaluated whether GXM could be used to generate MDSCs in vitro from bone marrow (BM) precursor cells. As positive controls, the addition of granulocyte-macrophage colony-stimulating factor (GM-CSF) alone or GM-CSF plus IL-6 combination to BM cell cultures induced the in vitro differentiation of MDSCs with increased levels of CD11b and GR1 (Fig. 2j and Supplementary Fig. 2g). Notably, the addition of GXM plus GM-CSF significantly increased the percentage of PMN-MDSCs (Fig. 2j and Supplementary Fig. 2g), which indicated that this combination was more potent in driving the in vitro differentiation of PMN-MDSCs. The addition of GXM plus GM-CSF significantly increased the expression of _Arg1_ (Fig. 2k). We also found that BM-MDSCs induced by GM-CSF plus IL-6 or GM-CSF plus GXM possessed tolerogenic activity, which was revealed by the ability to impair the proliferation of CD4$^+$ and CD8$^+$ T-cells (Fig. 2l and Supplementary Fig. 2h). Notably, treatment with an arginase inhibitor, Nω-hydroxy-nor-L-arginine (nor-NOHA), or supplementation with L-arginine significantly blocked the GXM-induced immunosuppressive activity of BM-MDSCs, which was evidenced by increased T-cell expansion (Fig. 2m and Supplementary Fig. 2i). These data confirmed that ARG1 was a key effector of the GXM-induced tolerogenic activity of MDSCs.

**CLEC2D specifically recognizes GXM from _C. neoformans_.** To examine whether PRRs expressed on MDSCs recognized GXM from _C. neoformans_, we performed RNA-sequencing analysis and found that GXM stimulation resulted in a cluster of upregulated genes encoding CLRs in BM-MDSCs (Fig. 3a). GXM stimulation induced an approximately tenfold increase in the gene expression of _Clec2d_, which was verified using quantitative real-time PCR (Fig. 3b). We generated soluble protein chimeras consisting of the C-type lectin-like domain (aa 61–191) of CLEC2D fused to the Fc region of human immunoglobulin IgG1 (hFc-CLEC2D, Supplementary Fig. 3a) and used hFc-CLEC2D as a probe to validate the recognition of _C. neoformans_ using flow cytometry. We identified that the C-type lectin-like domain of CLEC2D bound

encapsulated strain H99 but not the hypocapsular strain Cap59 (Fig. 3c), which suggested that GXM was exposed on the surface of _C. neoformans_ was a ligand of CLEC2D. We further performed an enzyme-linked immunosorbent assay (ELISA) to confirm the specific binding of CLEC2D with GXM (Fig. 3d). The binding to GXM was visualized using immunofluorescence microscopy, which revealed a uniform overlap of CLEC2D with GXM on the entire capsular surface of _C. neoformans_ (Fig. 3e).

GXM consists of an _O_-acetylated, −1-3 mannose backbone with xylose and glucuronic acid side chains[5], and CLEC2D binds some carbohydrates, including dextran sulfate and λ-carrageenan[31]. We found that pretreatment with mannan, dextran sulfate, or λ-carrageenan, but not xylose and dextran, completely blocked the binding of CLEC2D with _C. neoformans_ (Fig. 3f). We further generated soluble protein chimeras containing different C-type lectin-like fragments of CLEC2D (Supplementary Fig. 3b) and found that removal of aa 186–191 significantly impaired the binding of CLEC2D with _C. neoformans_ (Fig. 3g). However, fragments 60–140 and 60–99 of CLEC2D bound _C. neoformans_ (Fig. 3g). These data indicated that the C-type lectin-like domain of CLEC2D contained at least three binding sites for GXM.

_C. neoformans_ cells are highly negatively charged due to the presence of the capsule GXM[32], and a previous mutagenesis study of CLEC2D clearly revealed that mutations of positively charged residues, including Lys169, Arg175, Arg180, and Lys181, with glutamic acids, caused detrimental effects on binding, suggesting that charge distribution is critical for recognition by CLEC2D[33]. We generated soluble protein chimeras containing mutations of positively charged residues, including Lys169, Arg175, His176, Arg180, Lys181, Lys186, and His190, with glutamic acids in C-type lectin-like fragments of CLEC2D (Supplementary Fig. 3c). We found that mutations at Arg180, Lys181, and Lys186 completely impaired the binding of CLEC2D with _C. neoformans_ cells (Fig. 3h). These data confirmed the specific recognition of GXM by CLEC2D and indicated that positively charged residues of CLEC2D were essential for its binding with the negatively charged GXM.

_Clec2d_ **deficiency impairs MDSC activation to alleviate _C. neoformans_ infection.** To further determine whether depletion of _Clec2d_ impaired GXM-induced immunosuppressive activity of MDSCs, we constructed _Clec2d_-deficient mice with depletion of exons 2–5 using a CRISPR-Cas9-mediated guide RNA targeting approach (Supplementary Fig. 4a, b). PCR products from F0, F1, and subsequent offspring confirmed the presence of heterozygous and homozygous _Clec2d_ knockout (KO) mice (Supplementary Fig. 4c). We also found that CLEC2D protein was prominently expressed in the spleen and lung of mice, and we confirmed that _Clec2d_ was deficient in homozygous KO mice (Supplementary Fig. 4d). We performed RNA-sequencing analysis of BM-MDSCs

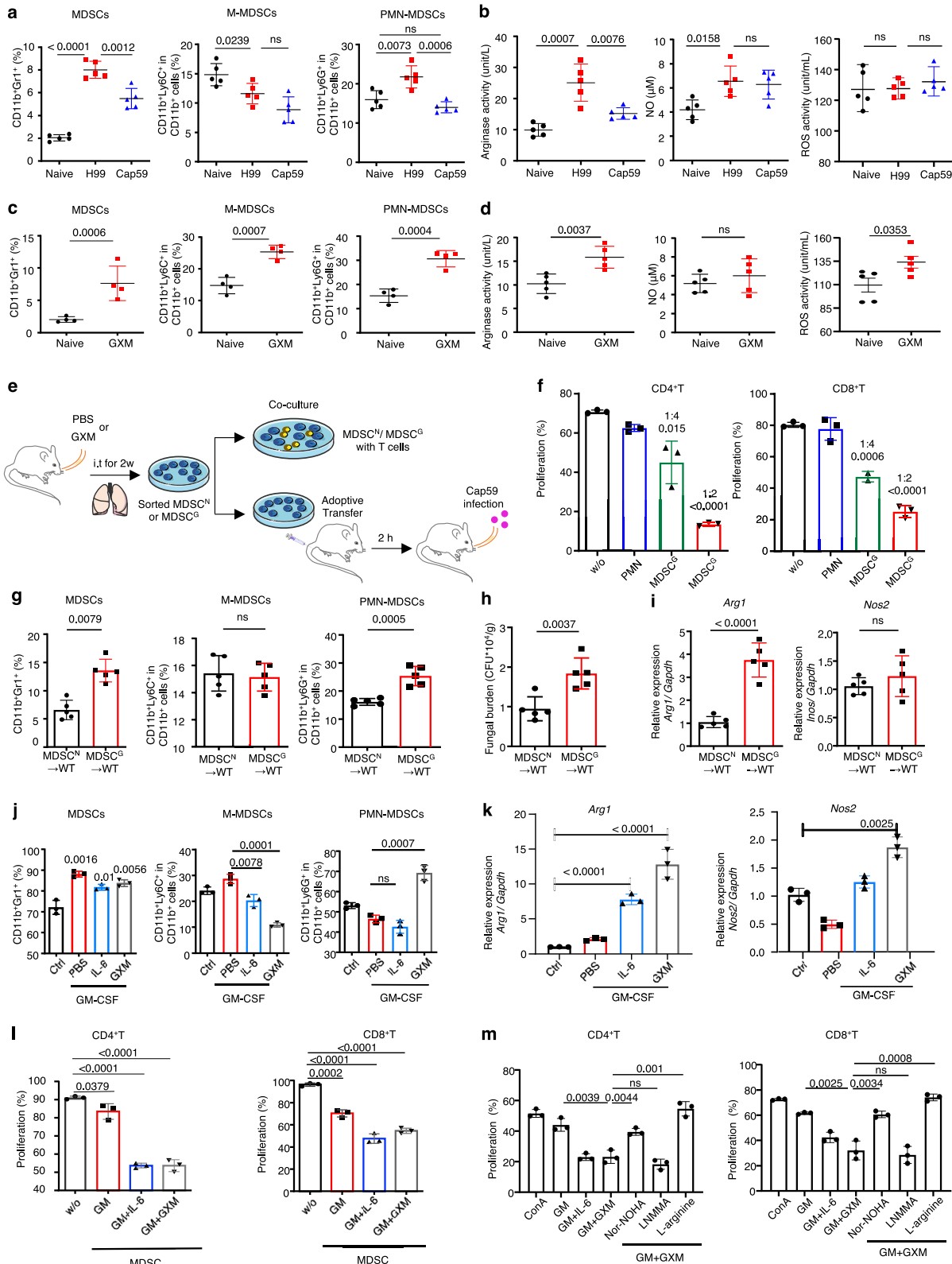

from WT and *Clec2d* KO mice and found that GM-CSF plus GXM stimulation resulted in 138 downregulated and 314 upregulated genes in *Clec2d*-deficient BM-MDSCs (Fig. 4a). The expression level of *Arg1* was significantly decreased in *Clec2d*-deficient BM-MDSCs (Fig. 4b).

We further determined whether depletion of *Clec2d* impaired the GXM-induced immunosuppressive activity of MDSCs in vitro and in vivo (Fig. 4c). We observed that *Clec2d* deficiency significantly blocked the immunosuppressive activity of BM-MDSCs induced by GM-CSF plus GXM, but not GM-CSF plus IL-6, which was evidenced by the ability to completely impair the upregulation of *Arg1* expression and significantly increase the proliferation of CD4$^+$ and CD8$^+$ T-cells by *Clec2d*-deficient BM-MDSCs (Fig. 4d, e and Supplementary Fig. 4e). These data

**Fig. 2 GXM induces the expansion and activation of MDSCs, especially PMN-MDSCs, to suppress T-cells and aggravate *C. neoformans* infection. a** The percentage of MDSCs, M-MDSCs, or PMN-MDSCs in mice, which were intratracheally infected with a single exposure of *C. neoformans* encapsulated strain H99 ($1 \times 10^3$ CFU/mouse) or hypocapsular strain *cap59Δ* mutant (Cap59, $1 \times 10^5$ CFU/mouse) for 14 days. **b** Activities of arginase, NO, and ROS in the lungs of H99- or Cap59-infected mice as shown in **a**. **c** The percentage of MDSCs, M-MDSCs, and PMN-MDSCs in mice treated without or with GXM (100 μg/mouse twice a week) for 14 days. **d** Activities of arginase, NO, and ROS of GXM-treated mice as shown in **c**. **e** Schematic strategy for sorting PMN from PBS-treated mice (PMN) or MDSCs from GXM-treated mice (MDSC^G), which were co-cultured with T-cells or adoptively transferred to wild-type mice ($1 \times 10^6$ cells/mouse). **f** Proliferation frequency of CD4$^+$ or CD8$^+$ T-cells, which were co-cultured for 72 h without (w/o) or with PMN sorted from PBS-treated mice (MDSC^N) or MDSCs sorted from GXM-treated mice (MDSC^G) as shown in **e**. **g–i** The percentages of MDSCs, M-MDSCs, and PMN-MDSCs (**g**), pulmonary fungal burden (**h**), or mRNA levels of *Arg1* and *Nos2* (**i**) in lungs of Cap59-infected wild-type mice (WT) for 14 days, which are adoptively transferred with MDSC^N or MDSC^G as shown in (**e**, $1 \times 10^6$ cells/mouse). **j** The percentage of bone marrow (BM)-derived MDSCs, M-MDSCs, and PMN-MDSCs, which were induced by medium (Ctrl), GM-CSF (GM, 40 ng/ml), GM-CSF + IL-6 (40 ng/ml), or GM-CSF + GXM (10 μg/well) for 5 days. **k** mRNA levels of *Arg1* and *Nos2* in BM-derived MDSCs generated as described in (**j**). **l** Proliferation frequency of CD4$^+$ or CD8$^+$ T-cells which were co-cultured for 72 h without (w/o) or with BM-derived MDSCs generated as described in (**j**). **m** Proliferation frequency of CD4$^+$ or CD8$^+$ T-cells, which were pretreated with nitric oxide synthase inhibitor L-NMMA, arginase inhibitor nor-NOHA, or L-arginine and then co-cultured for 72 h with BM-MDSCs generated by GM + GXM as described in **j**. Data were presented as mean ± SEM; $n = 3$ (**f**, **j–m**), $n = 4$ (**c**), or $n = 5$ (**a**, **b**, **d**, **g–i**) biologically independent samples. Data were analyzed by unpaired two-sided Student's *t*-test in **a–d**, **f–m**. Source data are provided as a Source Data file.

suggested that CLEC2D was critical for controlling the GXM-induced tolerogenic activity of MDSCs.

To evaluate the CLEC2D-mediated tolerogenic activity of MDSCs on the host defense against *C. neoformans* infection, we adoptively transferred WT and *Clec2d*-deficient BM-MDSCs induced by GM-CSF plus IL-6 or GM-CSF plus GXM to WT mice and monitored their impacts on fungal burdens in Cap59-infected mice (Fig. 4c). We found that the adoptive transfer of different MDSCs comparably increased the percentage of MDSCs in the lungs of Cap59-infected WT mice (Supplementary Fig. 4f). However, only adoptive transfer of *Clec2d*-deficient MDSCs induced by GM-CSF plus GXM dramatically decreased fungal burdens and *Arg1* expression in the lungs of Cap59-infected WT mice (Fig. 4f, g). We further adoptively transferred WT and Clec2d-deficient BM-MDSCs induced by GM-CSF plus GXM to *Clec2d*-deficient mice and monitored their impact in H99-infected mice (Fig. 4c). We also found that the adoptive transfer of different MDSCs comparably increased the percentage of MDSCs in the lungs of *Clec2d*-deficient mice after infection with H99 (Supplementary Fig. 4g). However, adoptive transfer of WT MDSCs induced by GM-CSF plus GXM significantly increased fungal burdens in the lungs of H99-infected *Clec2d*-deficient mice (Fig. 4h). Histological analysis of the infected lungs on day 14 showed that this transfer led to a significant increase in granulomatous lesions containing many yeasT-cells (Fig. 4i) and *Arg1* expression in lung tissue (Fig. 4i). Notably, adoptive transfer of WT MDSCs significantly impaired the frequency of IFN-γ-producing T$_H$1 cells and IL-17A-producing T$_H$17 cells in the lungs of H99-infected *Clec2d*-deficient mice (Fig. 4k). These data suggested that CLEC2D controlled the tolerogenic activity of MDSCs by recognizing GXM to aggravate *C. neoformans* infection.

**Inhibition of p38 activation is an effective immunotherapy against *C. neoformans* infection.** To determine the underlying mechanisms regulating the GXM-induced tolerogenic activity of MDSCs, we further analyzed the RNA-sequencing data of BM-MDSCs from WT and *Clec2d* KO mice and found that GM-CSF plus GXM stimulation resulted in the enrichment of MAPK, cAMP, and Rap1 signaling pathways (Fig. 5a). Gene set enrichment analysis (GSEA) suggested that the MAPK pathway was involved in regulating the GXM-induced tolerogenic activity of MDSCs when CLEC2D was engaged (Fig. 5b). We observed that *Clec2d* deficiency significantly impaired GXM-induced phosphorylation of p38 in BM-MDSCs (Fig. 5c). We further performed RNA-sequencing analysis of BM-MDSCs induced by GM-CSF plus GXM with or without SB202190 treatment, which is a specific inhibitor of p38, and found that

SB202190 treatment resulted in 175 downregulated and 81 upregulated genes (Supplementary Fig. 5a). The expression levels of *Arg1* and MAPK pathway-involved genes were significantly decreased in SB202190-treated BM-MDSCs (Fig. 5d and Supplementary Fig. 5b). We further confirmed that SB202190 treatment significantly blocked the immunosuppressive activity of BM-MDSCs induced by GM-CSF plus GXM, which was evidenced by the ability to completely impair the upregulation of *Arg1* expression and significantly increase the proliferation of CD4$^+$ and CD8$^+$ T-cells (Fig. 5e, f and Supplementary Fig. 5c). Inhibition of p38 by SB202190 did not influence the proliferation of CD4$^+$ or CD8$^+$T-cells when co-cultured with *Clec2d*-deficient BM-MDSCs (Fig. 5g and Supplementary Fig. 5d). These results suggested that the p38-MAPK pathway is a critical regulator of the GXM-induced suppressive function of MDSCs via the engagement of CLEC2D.

We further examined whether the inhibition of p38 activation in vivo may be used as immunotherapy against *C. neoformans* infection. We found that intraperitoneal injection of SB202190 significantly increased the survival rate of mice after infection with *C. neoformans* strain H99 (Fig. 5h). SB202190 treatment significantly decreased fungal burden and focal granulomatous lesions containing yeasT-cells in the lungs of H99-infected mice (Fig. 5I, j). This treatment significantly decreased the percentage of MDSCs, especially PMN-MDSCs, and the expression of *Arg1*, but not *Nos2*, in the lungs of H99-infected mice (Fig. 5k, l and Supplementary Fig. 5e). Therefore, SB202190 treatment significantly increased the frequency of IFN-γ-producing T$_H$1 cells and IL-17A-producing T$_H$17 cells in the lungs of H99-infected mice (Fig. 5m and Supplementary Fig. 5f). These data suggested that the inhibition of p38-mediated ARG1 production in MDSCs enhanced T-cell-mediated responses against *C. neoformans* infection.

**Vandetanib inhibits MDSC-derived arginase-1 production to enhance T-cell-based immunotherapy against *C. neoformans* infection.** To examine whether compounds developed for other therapeutic indications may be repurposed to inhibit p38-mediated ARG1 production in MDSCs, we screened 597 off-patent drugs and found that sorafenib, dasatinib, and vandetanib inhibited p38 phosphorylation in the immortalized MDSC cell line MSC-2 upon GXM stimulation (Supplementary Fig. 6a). However, only treatment with vandetanib significantly inhibited GXM-induced *Arg1* expression (Supplementary Fig. 6b). Vandetanib inhibited p38 phosphorylation in BM-MDSCs in a dose-dependent manner (Fig. 6a). Vandetanib is an oral inhibitor of vascular endothelial growth factor receptor 2

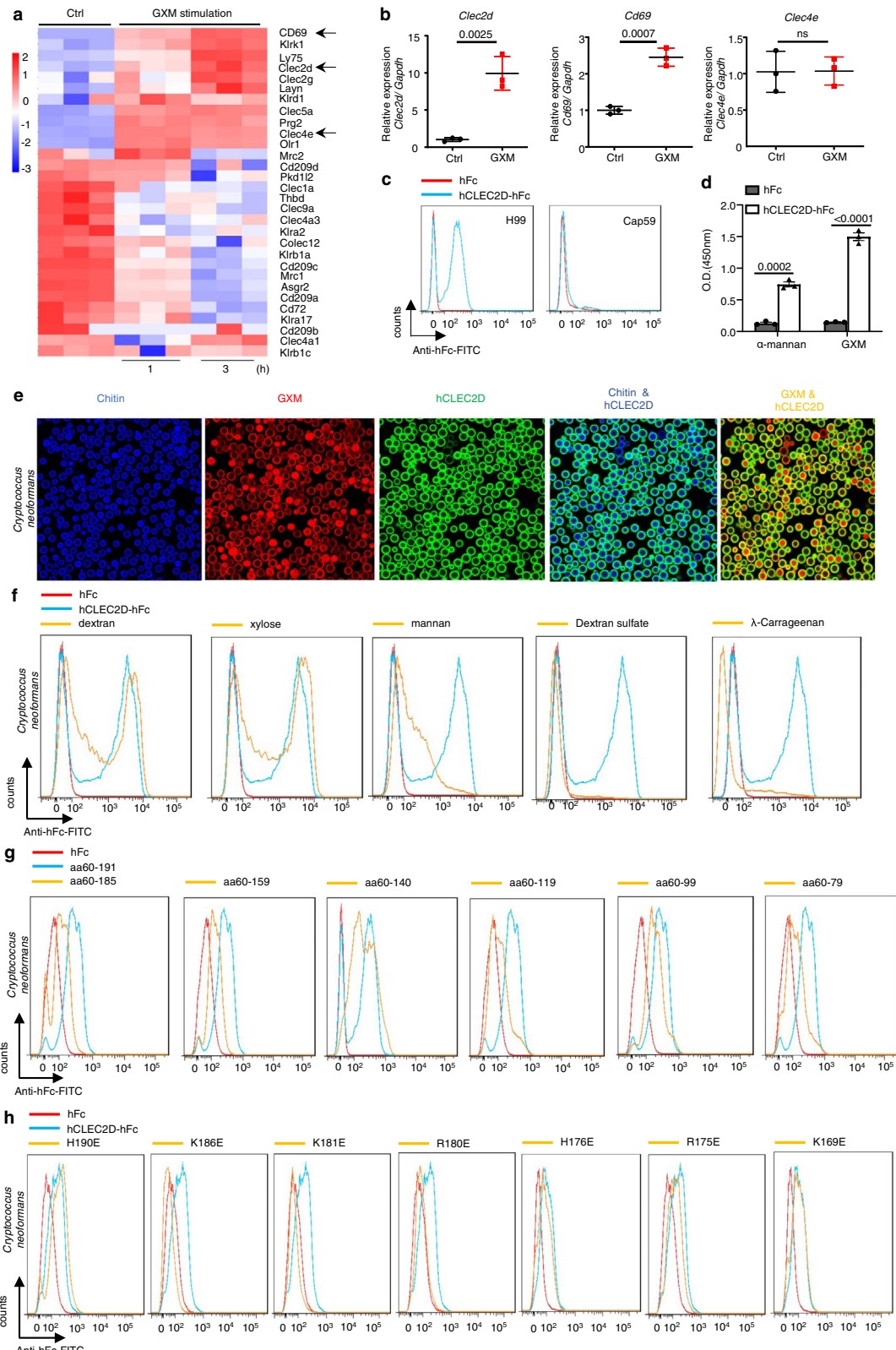

(VEGFR-2), epidermal growth factor receptor (EGFR), and Ret tyrosine kinases involved in tumor growth, progression, and angiogenesis[34]. We further performed RNA-sequencing analysis of BM-MDSCs induced by GM-CSF plus GXM with or without vandetanib treatment and found that this treatment resulted in 107 downregulated and 154 upregulated genes (Supplementary Fig. 6c). Seventeen upregulated and 39 downregulated genes in

BM-MDSCs were overlapped with SB202190 or vandetanib treatment (Fig. 6b, c). Notably, the expression levels of *Arg1* and MAPK pathway-involved genes were significantly decreased in BM-MDSCs treated with SB202190 or vandetanib (Fig. 6d and Supplementary Fig. 6d). We further confirmed that vandetanib treatment also significantly blocked the immunosuppressive activity of BM-MDSCs, which was evidenced by the ability to

**Fig. 3 CLEC2D specifically recognizes GXM from *C. neoformans*. a** Microarray analysis of differentially expressed genes encoding C-type lectin receptors in BM-derived MDSCs generated by GM-CSF (40 ng/ml) + GXM (10 μg/well), which were then treated with PBS (Ctrl) or GXM (10 μg/well) for the indicated time. **b** Quantitative analysis of *Cd69*, *Clec2d*, and *Clec4e* expression in BM-derived MDSCs prepared as shown in **a**. **c** Binding assay of the soluble protein chimera hCLEC2D-hIgG1-Fc or hIgG1-Fc (hFc) to *C. neoformans* using flow cytometry. **d** ELISA assay of the binding of hCLEC2D-hIgG1-Fc or hIgG1-Fc (hFc) with plate-coated GXM or α-mannan (1 μg/well). **e** Representative immunofluorescent staining assay of the binding of *C. neoformans* with hCLEC2D-hIgG1-Fc (anti-hIgG1-Fc as second Ab, green), Calcofluor White (50 μg/ml, Chitin staining, blue) or GXM. Chitin was stained with Calcofluor White (50 μg/ml) (blue), and 18B7 Ab (GXM, red). **f** Flow cytometry assay of the competitive binding of *C. neoformans* with CLEC2D-hIgG1-Fc or hIgG1-Fc (hFc) using the indicated polysaccharides. **g**, **h** Flow cytometry assay of the binding of *C. neoformans* with the indicated soluble protein truncations (**g**) or amino acid mutated proteins (**h**) of CLEC2D-hIgG1-Fc2. Data were presented as mean ± SEM, n = 3 (**b**, **d**) biologically independent samples. Data were analyzed by unpaired two-sided Student's *t*-test in **b** and one-way ANOVA adjusted for multiple comparisons in **d**. Source data are provided as a Source Data file.

---

significantly increase the proliferation of CD4$^+$ and CD8$^+$ T-cells (Fig. 6e and Supplementary Fig. 6e).

We found that intraperitoneal injection of vandetanib once every other day significantly increased the survival rate of mice after infection with *C. neoformans* strain H99 (Fig. 6f). Vandetanib treatment significantly decreased fungal burden and focal granulomatous lesions containing yeasT-cells in the lungs of H99-infected mice (Fig. 6g, h). This treatment significantly decreased the expression of *Arg1*, but not *Nos2*, in the lungs of H99-infected mice (Fig. 6i). Vandetanib treatment significantly increased the frequency of IFN-γ-producing $T_H1$ cells and IL-17A-producing $T_H17$ cells and the amount of IFN-γ and IL-17A in the lungs of H99-infected mice (Fig. 6j, k and Supplementary Fig. 6f, g). Therefore, these data suggested that vandetanib inhibited MDSC-derived ARG1 production to enhance T-cell-based immunotherapy against *C. neoformans* infection.

## Discussion

*C. neoformans* infection is primarily controlled by cell-mediated immunity in immunocompetent patients who develop a CD4$^+$T$_H$1 response[35]. It has been well-documented that *C. neoformans* strain encapsulated with GXM are able to inhibit the development of T$_H$1-mediated protective response through the induction of IL-10[36]. Beside this, GXM also inhibits the efficiency of the antigen presentation process of macrophages probably through downregulating major histocompatibility complex (MHC) class II and B7 expression[16,17], which leads to an inhibition of T-cell proliferation in response to *C. neoformans*[14]. In addition to having potent immunosuppressive effects on cytokine production and co-stimulation molecular expression, GXM is also capable of inducing apoptosis of T-cells by inducing Fas-L expression in macrophages[11,12]. In the present study, we showed that GXM played a dominant role in inducing the accumulation and activation of MDSCs, especially PMN-MDSCs, which are well-characterized by their capacity to suppress T-cell responses[25]. Notably, an accumulation and immunosuppressive activity of MDSCs were shown after infection with *M. tuberculosis*[19–21], hepatitis C virus[22,23] and parasites[24]. However, the role of MDSCs in fungal infection was rarely reported. In a recent study, Rieber N et al. show that β-glucans exposed to *C. albicans* can induce PMN-MDSCs through Dectin-1 receptor and its downstream adapter protein CARD9 and that PMN-MDSCs were protective from systemic *C. albicans* infection by suppressing hyperinflammatory responses[37]. However, we showed that GXM from *C. neoformans* induced the tolerogenic activity of neutrophilic MDSCs to aggravate their infections. Depletion of PMN-MDSCs using anti-Ly6G or anti-DR5 antibodies significantly enhanced host defense against *C. neoformans* infection. More importantly, we identified a CLR, CLEC2D, expressed on MDSCs to recognize GXM for controlling the tolerogenic activity of MDSCs both in vitro and in vivo.

The roles of CLRs, including Dectin-1, Dectin-2, and Dectin-3, in protection against cryptococcosis remain controversial. Dectin-1

is redundant for protection against *C. neoformans* serotype A[38,39]. Dectin-2 plays a minor role during infection with *C. neoformans* serotype D, but there are slight alterations in the increased T$_H$2-cell-mediated responses[40]. Our previous study demonstrated that Dectin-3 recognized GXM from *C. neoformans* serotype AD and *C. gattii* serotype B to initiate host defense against pulmonary infection[30]. However, Dectin-3 was not required for host defense against pulmonary infection with *C. neoformans* serotype A[30,41]. Our present study showed that CLEC2D specifically bound to GXM from *C. neoformans* serotype A. We further showed that positively charged residues, including Arg180, Lys181 and Lys186, of CLEC2D were essential for its binding with the negatively charged GXM. A previous mutagenesis study clearly revealed that the mutations of the positively charged residues (Lys169, Arg175, Arg180, and Lys181) of CLEC2D with glutamic acids causes detrimental effects on the binding, suggesting that the charge distribution is critical for the recognition[33]. CLEC2D is a C-type lectin-like protein that was first characterized as the ligand for the CD161 receptor that primarily expressed on NK cells[42]. A previous study showed that CLEC2D functioned as a CLR to bind some carbohydrates, including sulfated glycosaminoglycans, such as dextran sulfate, fucoidan, and λ-carrageenan[31]. A recent study has shown that CLEC2D binds to histones that are released upon necrotic cell death to regulate inflammation and tissue damage[43]. However, CLEC2D lacks ITAM/ITIM motifs or known signaling adapters used by other stimulatory CLRs. A recent study also failed to detect any CLEC2D-dependent stimulation of multiple MAP kinases, Syk and NF-κB by purified histones in macrophages[43]. Our data showed that CLEC2D recognized GXM from *C. neoformans* to mediate the phosphorylation of p38, which regulated *Arg1* expression in MDSCs. More importantly, we showed that adoptive transfer of *Clec2d*-deficient MDSCs induced by GM-CSF plus GXM into wild-type mice significantly decreased their pulmonary fungi burden and *Arg1* expression after infection with *C. neoformans* hypocapsular strain Cap59. In contrast, adoptive transfer of wild-type MDSCs induced by GM-CSF plus GXM into *Clec2d*-deficient mice significantly increased their pulmonary fungi burden and *Arg1* expression to suppress the function of IFN-γ-producing T$_H$1 cells and IL-17A-producing T$_H$17 cells after infection with *C. neoformans* encapsulated strain H99. These data suggested that CLEC2D partially controlled the tolerogenic activity of MDSCs through recognizing GXM to aggravate *C. neoformans* infection. Further studies are required to determine whether other receptors are involved in the recognition of GXM from *C. neoformans* to affect the suppressive activities of MDSCs.

It has been well-documented that MDSCs are one of the primary immunosuppressive cells acting as an escape mechanism for cancer cells by inhibiting T-cell activity. However, the identity of MDSCs is highly controversial. MDSCs can be broadly classified into two groups: polymorphonuclear (PMN-MDSC) and monocytic (M-MDSC). In mice, M-MDSCs and PMN-MDSCs are defined as CD11b$^+$Ly6C$^{high}$Ly6G$^-$ and CD11b$^+$ Ly6C$^{low}$Ly6G$^+$,

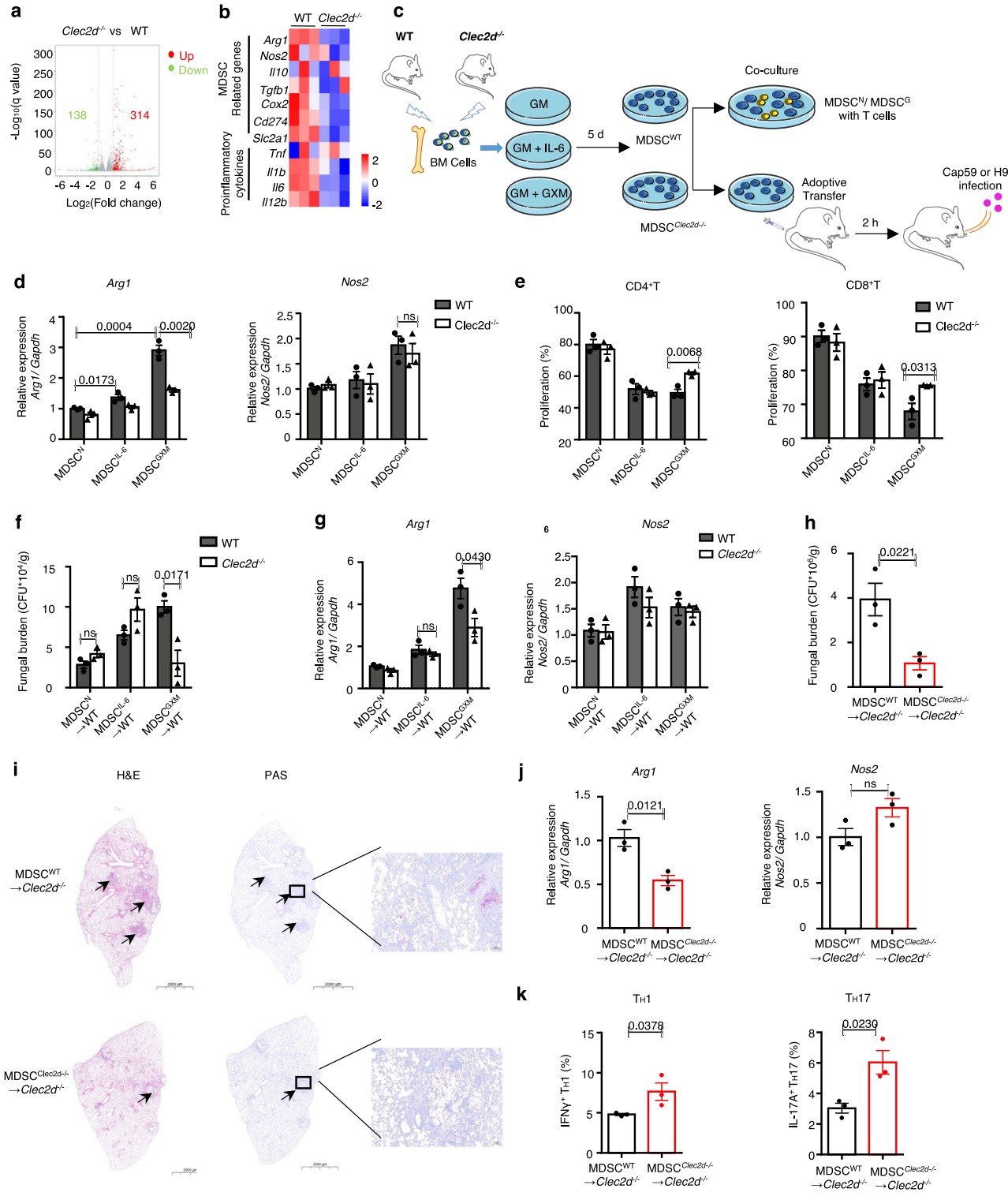

respectively. A recent study has suggested that PMN-MDSCs may be distinguished from neutrophils by surface markers CD244 and CD115[27]. In contrast, human MDSCs are defined based on myeloid cell markers (CD11b+CD33+HLA-DR−Lin−) and the same two MDSC subsets can be characterized by CD11b +CD33+HLA−DR−CD14+CD15− for M-MDSC and CD11b +CD33+HLA−DR−CD14−CD15+ for PMN-MDSC. Our present study showed that infection with *C. neoformans* strain encapsulated with GXM induced the recruitment of MDSCs, especially PMN-MDSCs, in mice. More importantly, we observed that non-

HIV 15 patients with pulmonary cryptococcosis and 14 patients with cryptococcal meningitis (Supplementary Table 1) significantly accumulated PMN-MDSCs, but not M-MDSCs, in PBMCs compared to healthy controls. However, it remains unclear whether HIV infection affect the accumulation of MDSCs in patients with pulmonary cryptococcosis or cryptococcal meningitis.

Accumulating research indicates that inhibiting the potent immunosuppressive mechanisms of MDSCs can be a therapeutic target to re-establish T-cell activity and immunotherapy success[44].

**Fig. 4 Clec2d deficiency impairs MDSC activation to alleviate *C. neoformans* infection. a** 138 downregulated (green) and 314 upregulated genes (red) in BM-derived MDSCs from wild-type (WT) or *Clec2d*[−/−] mice, which were generated by GM-CSF (GM, 40 ng/ml) + GXM (10 µg/well) for 5 days and then treated with GXM (10 µg/well) for 3 h. The volcano plot showed differentially expressed genes based on absolute fold change >2 and FDR <0.05. **b** Heatmap of differentially expressed MDSC-related and pro-inflammatory genes as shown in **a**. **c** Schematic strategy for generating wild-type and *Clec2d*-deficient BM-derived MDSCs (MDSC[WT] and MDSC[Clec2d−/−]), which were co-cultured with T-cells or adoptively transferred to wild-type or *Clec2d*-deficient mice (1 × 10⁶ cells/mouse). **d** mRNA levels of *Arg1* and *Nos2* in wild-type and *Clec2d*-deficient BM-derived MDSCs, which were generated with GM-CSF (GM, 40 ng/ml, MDSC[N]), GM + IL-6 (40 ng/ml, MDSC[IL-6]), or GM + GXM (10 µg/well, MDSC[GXM]) for 5 days. **e** Proliferation frequency of CD4⁺ or CD8⁺ T-cells, which were co-cultured for 72 h with MDSC[N], MDSC[IL-6], or MDSC[GXM] as shown in **d**. **f, g** Fungal burden (**f**) or mRNA levels of Arg1 and Nos2 (**g**) in lungs of Cap59-infected mice (1 × 10⁵ CFU/mouse) on day 14, which are adoptively transferred with MDSC[N], MDSC[IL-6], or MDSC[GXM] (1 × 10⁶ cells/mouse) as shown in **d**. **h–k** Fungal burden (**h**), representative histological images (**i**), mRNA levels of Arg1 and Nos2 (**j**), or the frequency of IFNγ⁺ Th1 and IL-17A⁺ Th17 (**k**) in lungs of *C. neoformans* strain H99-infected *Clec2d*⁻ᐟ⁻ mice (1 × 10³ CFU/mouse) on Day 14, which are adoptively transferred with MDSC[WT] and MDSC[Clec2d-/-] (1 × 10⁶ cells/mouse) generated by GM (40 ng/ml) + GXM (10 µg/well). Scale bars = 2000 µm, arrows indicate yeasT-cells in granulomatous lesions. Data were presented as mean ± SEM, n = 3 (**d–h**, **j**, **k**) biologically independent samples. Data were analyzed by unpaired two-sided Student's *t*-test in **e**, **f–h**, **j**, **k** and one-way ANOVA adjusted for multiple comparisons in **d**. Source data are provided as a Source Data file.

Inhibition of cyclooxygenase-2 (COX-2) with celecoxib has been successful in repressing MDSC-derived *Arg1* expression and ROS production, thereby enhancing anti-cancer immune responses[45]. Administration of Phosphodiesterase-5 (PDE-5) inhibitors, such as sildenafil and tadalafil, has been reported to abrogate MDSC immunosuppressive mechanisms through repressing MDSC-derived *Arg1* and *Nos2* expression, thereby enhancing T-cell activity and prolonging survival in vivo[46,47]. Clinical trials with PDE-5 inhibitors have also shown positive results in patients with head and neck squamous cell carcinoma and metastatic melanoma[48]. Our present study showed that pharmacological inhibition of MDSC-derived *Arg1* expression by either SB202190, a specific inhibitor of p38, or vandetanib, an orally available receptor tyrosine kinase inhibitor, significantly enhanced T-cell-mediated antifungal responses against *C. neoformans* infection. However, it needs further studies to explore whether enhancing the function of T-cells through inhibiting MDSC-derived ARG1 production could reduce the mortality and morbidity of AIDS-associated cryptococcal meningitis since these patients have very few CD4⁺ T-cells and may not therefore be able to respond to our proposed T-cell mediated immunotherapy.

We showed that GXM was sensed by the host via CLEC2D to induce the immunosuppressive activities of MDSCs, which inhibited antifungal immunity against *C. neoformans* infections. We further showed that inhibition of MDSC-derived ARG1 production could be used as an immunotherapy strategy against *C. neoformans* infection. A greater understanding of how MDSCs mediate responses to fungal pathogens offers great promise for the development of new immunotherapies, which are desperately needed to combat these devastating infections.

## Methods

**Ethics statement**. All animal experiments were performed according to the protocol approved by the Animal Ethics Committee of Tongji University School of Medicine (protocol No. TJAA09021101). Studies of human PBMCs were approved by the Human Research Committee of Tongji University School of Medicine (protocol No. 2021TJDX019) and Shanghai Jiaotong University School of Medicine (protocol No. KY2020-334).

**Human subjects**. Peripheral blood samples from healthy control individuals (HC, n = 6), patients with pulmonary cryptococcosis (PM, n = 15), or cryptococcal meningitis (CM, n = 14) were collected at Shanghai Huashan Hospital, Fudan University School of Medicine (Shanghai, China). The diagnosis of pulmonary cryptococcosis or cryptococcal meningitis was based on a combination of clinical and radiological suspicion together with laboratory confirmation[49], and the detailed characteristics of patients are shown in Supplementary Table 1. About 10⁵ PBMC cells were collected by flow cytometry, gating strategy for human PBMC assay was shown in Supplementary Fig. 1e. The percentages of MDSC, M-MDSCs, and PMN-MDSCs was counted in 10⁵ PBMCs. All participants were informed both orally and in writing of potential risks, and discomforts associated with participation before written consent was obtained.

***C. neoformans* strains and culture conditions**. *C. neoformans* strain H99 and mutant strain *ΔCap59* were gifted from the Shanghai Institute of Fungal Medicine, China. The yeasT-cells were stored at −80 °C and reactivated on Sabouraud dextrose agar (SDA) plates before use. Strains were cultured in yeast extract peptone dextrose (YPD) medium at 30 °C for 16 h before infecting mice.

**Mice**. C57BL/6 mice were purchased from Vital River (Shanghai, China). Six- to eight-week-old female mice were used in experiments. All mice were bred under specific pathogen-free conditions in ventilated cages with ad libitum food and water at Tongji University School of Medicine (Shanghai, China). Housing conditions were as following: dark/light cycle 12/12 h, ambient temperature around 21–22 °C and humidity between 40 and 70%. The *Clec2d* knockout mouse line was generated by using CRISPR-Cas9 methods[50]. The knockout line was generated with four single-guide RNA (sgRNA) target sequences, which were shown in Supplementary Table 2. The depletion of exons 2 and 5 results in knockout of *Clec2d* gene (Supplementary Fig. 4a). Cas9 mRNA and gRNA were microinjected into the fertilized eggs of C57BL/6 J mice to obtain F₀ generation mice. F₀ founders were crossed with C57BL/6 mice to obtain F₁ offspring. The knockout mice used in this study were obtained by backcrossing F₁ generation to C57BL/6 mice for more than ten generations. All animal studies were performed with sex- and age-matched mice after approval from the Institutional Laboratory Animal Care and Use Committee (protocol No. TJAA09021101).

**Mouse models**. Seven-week-old female C57BL/6 (WT) or *Clec2d*⁻ᐟ⁻ mice were used for establishing *C. neoformans* infection model[30]. Mice were intratracheally challenged with *C. neoformans* strain H99 (1 × 10³ CFU/mice) or Cap59 (1 × 10⁵ CFU/mice) in 35 µL PBS and sacrificed on Day 14, lung tissues were extracted for subsequent experiments. To construct GXM-treated inflammation model, 100 µg GXM was dissolved in 35 µL PBS and intratracheally injected into mice twice a week and sacrificed on Day 14. To test the therapeutic efficacy of vandetanib, H99-infected mice were treated orally with vehicle (10% DMSO + 40% PEG-300 + 10% Tween-80 + 40% H₂O) or vehicle + vandetanib (50 mg/kg) every other day. On Day 14 post-infection, mice were sacrificed, and lung tissues were extracted for analysis by quantitative real-time PCR, histopathology, and flow cytometry. On Day 21, a fungal burden in the lung, spleen, and brain tissues was measured.

**Cell line**. The immortalized MDSC cell line MSC-2 was gifted by Professor Zhihai Qin at the Institute of Biophysics, Chinese Academy of Sciences. The cells were passaged in Dulbecco's modified Eagle's medium (HyClone Laboratories) containing 10% fetal bovine serum, 100 U/mL penicillin, and 100 µg/mL streptomycin. Cell line was authenticated by sinobiological company (Beijing, China). The cell line is tested regularly by using GMyc-PCR Mycoplasma Test Kit (Yeasen, 40601ES10) and has no mycoplasma contamination.

**Fungal burden analysis**. Six-week-old female C57BL/6 (WT) or *Clec2d*⁻ᐟ⁻ *C. neoformans* infected mice or GXM pretreated mice were sacrificed on Day 14, and the lungs were dissected, excised, and homogenized sufficiently in 1 mL PBS by teasing with a stainless mesh at room temperature. The homogenates were diluted appropriately with PBS, inoculated with 100 µL PBS on SDA plates and cultured at 30 °C for 2 days, and the resulting colonies were counted.

**Adoptive transfer experiments**. CD11b⁺Gr1⁺ cells from 7-week-old female C57BL/6 (WT) or *Clec2d*⁻ᐟ⁻ mice were sorted from naïve mice or GXM-treated mice in which 100 µg GXM was dissolved in 35 µL PBS and intratracheally injected into mice twice a week and sacrificed on Day 14. Two hours after intravenous adoptive transfer, mice were challenged with 1 × 10⁵ CFU of *C.*

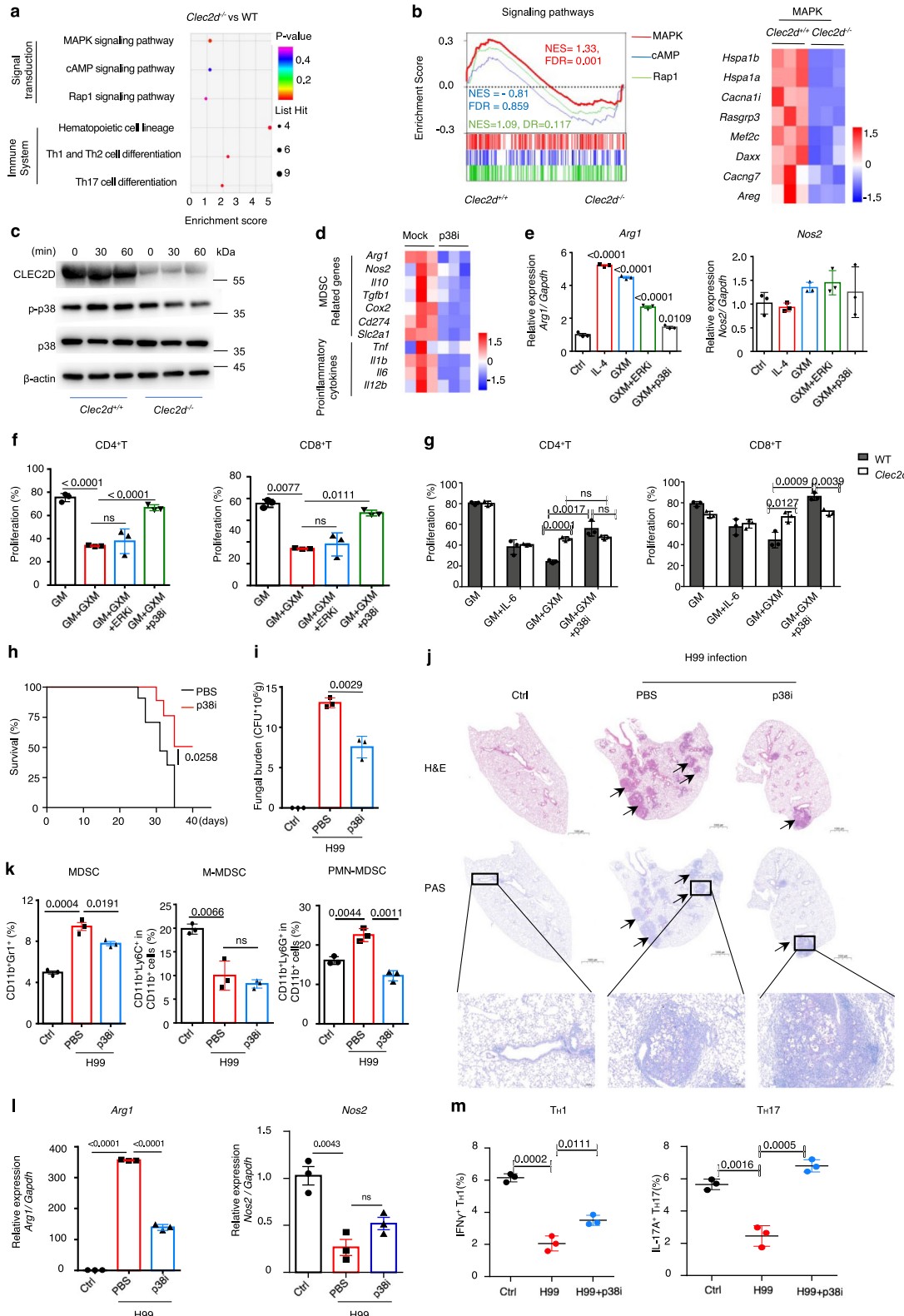

neoformans strain Cap59. On Day 14 after infection, mice were scarified for performing subsequent experiments.

For ex vivo adoptive transfer experiments, myeloid cells freshly isolated from bone marrow were pretreated with GM-CSF, GM-CSF + IL-6, or GM-CSF + GXM for 5 days, then collected and intravenously transferred into WT or $Clec2d^{-/-}$ mice ($1 \times 10^6$ cells in 200 μL PBS) as described above. Two hours after adoptive transfer, mice were challenged with $1 \times 10^5$ CFU of C. neoformans strain H99. On Day 14 after infection, mice were scarified for performing subsequent experiments.

**Expression of fusion protein and immunolabeling.** The expression of soluble fusion proteins was described as previously[51,52]. Briefly, the gene fragment of human CLEC2D C-type lectin-like domain (CTLD) was amplified from human PBMCs total cDNA and then cloned into the pFUSE-hIgG1-Fc2 expression vector (Invivogen, Catalog # pfuse-hg1fc2, Version # 06G05-MT). Sequenced plasmid was transfected into HEK293T-cells using Lipo2000 (Thermo Fisher Scientific, 11668019) according to the manufacturer's instructions. The soluble fusion protein was purified from the supernatants by dialysis against water and enriched by high

**Fig. 5 Inhibition of p38 activation is an effective immunotherapy against *C. neoformans* infection. a** KEGG analysis of differentially expressed genes involved in the signal transduction or the immune system in BM-derived MDSCs from wild-type (WT) or *Clec2d$^{-/-}$* mice, which were generated by GM-CSF (GM, 40 ng/ml) + GXM (10 μg/well) and then treated with GXM (10 μg/well) for 3 h. **b** Gene set enrichment analysis (GSEA) plots (Right) of the indicated signaling and Heatmap (Left) of differentially expressed MAPK-related genes in BM-derived MDSCs from WT or *Clec2d$^{-/-}$* mice as shown in **a**. *P* values were calculated by hypergeometric test and adjusted for multiple comparisons. **c** Western blot assay of the indicated proteins in BM-derived MDSCs from WT or *Clec2d$^{-/-}$* mice, which were generated by GM (40 ng/ml) + GXM (10 μg/well) and then treated with GXM (10 μg/well) for the indicated time. Representative blots were shown from three independent experiments. **d** Heatmap of differentially expressed MDSC-related and pro-inflammatory genes in wild-type BM-derived MDSCs, which were generated by GM (40 ng/ml) + GXM (10 μg/well) and then treated with GXM (10 μg/well) combined with or without p38 inhibitor SB202190 (p38i, 10 μM) for 3 h. **e** mRNA levels of *Arg1* and *Nos2* in cell line MSC-2, which were treated with PBS (control), IL-4 (10 ng/ml), GXM (10 μg/well), GXM + p38 inhibitor SB202190 (p38i, 10 μM), or GXM + ERK1/2 inhibitor U0126 (ERKi, 10 μM) for 6 h. **f** Proliferation frequency of CD4$^+$ and CD8$^+$ T-cells, which were co-cultured for 72 h with MSC-2 cells pretreated as shown in **e**. **g** Proliferation frequency of CD4$^+$ and CD8$^+$ T-cells, which were co-cultured with wild-type (WT) and *Clec2d$^{-/-}$* BM-derived MDSCs with or without treatment of p38 inhibitor SB202190 (p38i, 10 μM) for 72 h. **h** Survival assay of *C. neoformans* strain H99-infected mice (*n* = 8, 1 × 10$^3$ CFU/mouse), which were intraperitoneally treated with or without p38 inhibitor SB202190 (p38i, 25 μg/kg every other day). **i–m** Fungal burden (**i**), representative histological images with hematoxylin-eosin (H&E) and Periodic Acid-Schiff (PAS) staining (**j**), the percentage of MDSCs, M-MDSCs, and PMN-MDSCs (**k**), mRNA levels of *Arg1* and *Nos2* (**l**), and the frequency of IFNγ$^+$ T$_H$1 and IL-17A$^+$ T$_H$17 cells (**m**) in the lungs of *C. neoformans* strain H99-infected mice (1 × 10$^3$ CFU/mouse) on Day 14, which were intraperitoneally treated with or without p38 inhibitor SB202190 (p38i, 25 μg/kg every other day). Scale bars = 1000 μm; arrows indicate yeasT-cells in granulomatous lesions. Data were presented as mean ± SEM, *n* = 3 (**e–g**, **l**, **k–m**), *n* = 8 (**h**) biologically independent samples. Data were analyzed by unpaired two-sided Student's *t*-test in **e**, **f**, **i**, **k–m**, one-way ANOVA adjusted for multiple comparisons in **g** and two-sided log-rank (Mantel–Cox) tests in **h**. Source data are provided as a Source Data file.

concentration of PEG2000 solution. All reagent information and primers used in this study is listed in Supplementary Table 3.

For ELISA assay, 1 μg GXM or α-mannan was dissolved in 100 μL PBS and coated in 96-well plate overnight. The 96-well plate were washed three times with PBST (PBS + 0.5% Tween 20) and successively incubated with 100 μL purified fusion protein and HRP-labeled goat anti-human IgG1-Fc antibody. After washing with PBST, the plate was added TMB to incubate for 15 min, 1 M H$_3$PO$_4$ was used to terminate the reaction and the optical density (OD) at 450 nm was measured by microplate reader (Multiskan FC, Thermo Fisher Scientific).

For flow cytometry assay, *C. neoformans* cells were incubated in 100 μL purified fusion protein with 1 mg glucuronic acid at 37°C for 1 h, after twice wash with PBS (containing 1% FBS), *C. neoformans* cells were added 100 ul FITC-labeled goat anti-human IgG1-Fc antibody (Jackson ImmunoResearch, 109-095-190, 1:100) to incubate for 30 min at 4 °C, finally, washed twice with PBS, and analyzed by Flow analyzer (BD FACSVerse).

For microscopy assay, *C. neoformans* yeasT-cells were incubated with fusion proteins at 37 °C for 1 h. After wash with PBS (containing 1% FBS) for three times, *C. neoformans* yeasT-cells were incubated with murine monoclonal antibody 18B7 (Sigma-Aldrich, MABF2069,1 μg/mL) for 30 min. Then, cells were incubated for 30 min with secondary antibodies, which included Alexa Fluor 594-labeled donkey anti-mouse IgG (Abcam, ab150116, 1:200) and a FITC-labeled goat anti-human Fc fragment (Jackson Immuno Research, 109-007-008, 1:50). On the other hand, *C. neoformans* yeasT-cells were incubated with Calcofluor White (Sigma-Aldrich, 18909, 50 μg/mL) for 20 min to stain chitin. *C. neoformans* yeasT-cells were observed using fluorescence microscopy (ZEISS LSM880, Germany).

**Flow cytometry assay**. Single-cell suspensions of peripheral blood or these prepared from lung tissue were stained with the following directly labeled antibodies: PerCP-Cy5.5-anti-mouse CD11b (Clone M1/70, BD Pharmingen, 561114, 1: 100); APC-anti-mouse Gr1(Clone RB6-8C5, BD Pharmingen, 553129, 1: 100); BV421-anti-mouse Ly6G (Clone 1A8, Biolegend 127628, 1: 100); FITC-anti-mouse Ly6G (Clone 1A8, eBioscience, 11-9668-82, 1:100); PE- anti-mouse Ly6C (Clone HK1.4, Biolegend, 128008, 1:100); APC-anti-mouse CD8 (Clone 53-6.7, Biolegend, 100712, 1:100); BV421-anti-mouse CD4 (Clone GK1.5, Biolegend, 100438, 1:100); APC-anti-mouse IFNγ (Clone XMG1.2, Biolegend, 505810, 1:100); PE-anti-mouse IL-17A (Clone ebio17B7, Biolegend, 559502). Detailed antibody information was summarized in Supplementary Table 3. Stained cells were collected and analyzed by FACSVerse device (BD Biosciences) and data was analyzed by Flowjo v10.

**Western blot analysis**. Cells (2 × 10$^6$) were harvested and lysed by RIPA buffer. In brief, cell extracts were collected and stored at −80 °C. Proteins were separated by homogeneous SDS-PAGE and transferred to a nitrocellulose membrane (pore size 0.45 μm; Merck Millipore). Primary antibodies which were used in this study: phospho-p38 (D3F9, Cell Signaling Technology, 4511 S, 1:1000); p38 (D13E1, Cell Signaling Technology, 8690 S, 1:1000); phospho-ERK (D13.14.4E, Cell Signaling Technology, 4370 S, 1:1000); ERK (137F5, Cell Signaling Technology, 4695 S, 1:1000); GAPDH (D16H11, Cell Signaling Technology, 5174 S, 1:1000); CLEC2D monoclonal antibody was generated by our lab (Clone 2260CT10.1.3.2), western blot validation of extracts from primary bone marrow cells. Primary antibody binding was visualized by chemiluminescence using HRP-conjugated goat anti-mouse (1:5000) or goat anti-rabbit IgG secondary antibodies (1:5000; both Cell Signaling Technology).

**Quantitative real-time PCR**. Total RNA was extracted from cells (1 × 10$^5$) using TRIzol (Thermo Fisher Scientific, 10296010), and cDNA was synthesized using EasyScript First-Strand cDNA Synthesis SuperMix (TIANGEN, KR106-02). Primers used in this study were shown in Supplementary Table 2. Individual mRNA quantities were quantified from Ct values in comparison with standard curves and calculated in relation to *Gapdh* as housekeeping genes. Expression level refers to the fold change of relative expression level of a treated sample in comparison with their respective controls.

**Primary MDSCs sorting**. Seven-week-old female C57BL/6 (WT) or *Clec2d$^{-/-}$* mice were intratracheally challenged with GXM (100 μg/mice, twice a week) for 14 days. Lungs were separated and incubated with 0.1% collagenase type 2 (Worthington Biochemical, Lakewood, NJ) in RPMI 1640 medium (HyClone Laboratories) for 2 h at 37 °C to obtain single-cell suspensions. Single-cell suspensions were fractionated using Percoll (GE Healthcare, Uppsala, Sweden) density gradient centrifugation then stained with directly labeled antibodies (PerCP-Cy5.5-anti-mouse CD11b and APC-anti-mouse Gr1). CD11b$^+$Gr1$^+$ cells were subsequently isolated using flow cytometric FACSAria II (BD Biosciences, San Jose, Calif).

**PMN-MDSC depletion**. To deplete PMN-MDSC, 6-week-old female C57BL/6 infected mice were injected intraperitoneally with anti-Ly6G antibody (1A8, BioX Cell, BE0075, 200 mg/mice), anti-DR5 antibody (MD5-1, BioX Cell, BE0161, 200 mg/mice), or anti-IgG (TNP6A7, BioX Cell, BP0290, 200 mg/mice) twice a week (*n* = 3 for each group). On Day 14 post-infection, lungs or peripheral blood samples were obtained for performing subsequent experiments.

**In vitro generation of BM-derived MDSCs**. Bone marrow cells were obtained from the femurs and tibias of 8-week-old female C57BL/6 mice and cultured in six-well plates in RPMI 1640 medium containing 10% FBS. The medium was supplemented with 40 ng/mL GM-CSF (PeproTech, AF-315-03,), GM-CSF + 40 ng/mL IL-6 (PeproTech, 216-16-100) or GM-CSF + GXM (10 μg/well). Vandetanib (Selleck, ZD6474) was added at a final concentration of 2 μM in the drug treatment group, and DMSO was used for the control group. 5 days later, the cells were harvested, and the proportion and suppressive function of MDSCs were analyzed using flow cytometry.

**T-cell proliferation suppression assays**. T-cell proliferation suppressive activities were measured by using bone marrow-derived MDSCs or MSC-2. Splenocytes were isolated from 8-week-old female C57BL/6 mice and stained with 1 mmol/L 5(6)-carboxyfluorescein diacetate succinimidyl ester (CFSE; Thermo Fisher Scientific) according to the manufacturer's instructions. CFSE-labeled splenic T-cells were seeded in 96-well plates at 3 × 10$^5$ cells/well and co-cultured with MDSCs (ratio = 2:1 or 1:1) in the presence of 1 μg/mL ConA for 72 h. The suppressive function of T-cell proliferation was determined using flow cytometry. nor-NOHA (10 mmol/L, Cayman Chemical, FY28418-5), L-NMMA (10 mmol/L, Cayman Chemical, FY10005031-50), or L-arginine (1 mmol/L, Sigma-Aldrich, A8094) were used to determine their inhibition on the suppressive function of MDSCs.

**Monoclonal antibody preparation**. The generation of monoclonal antibodies (mAb) to CLEC2D was performed as follows. Six-week-old female C57BL/6 mice

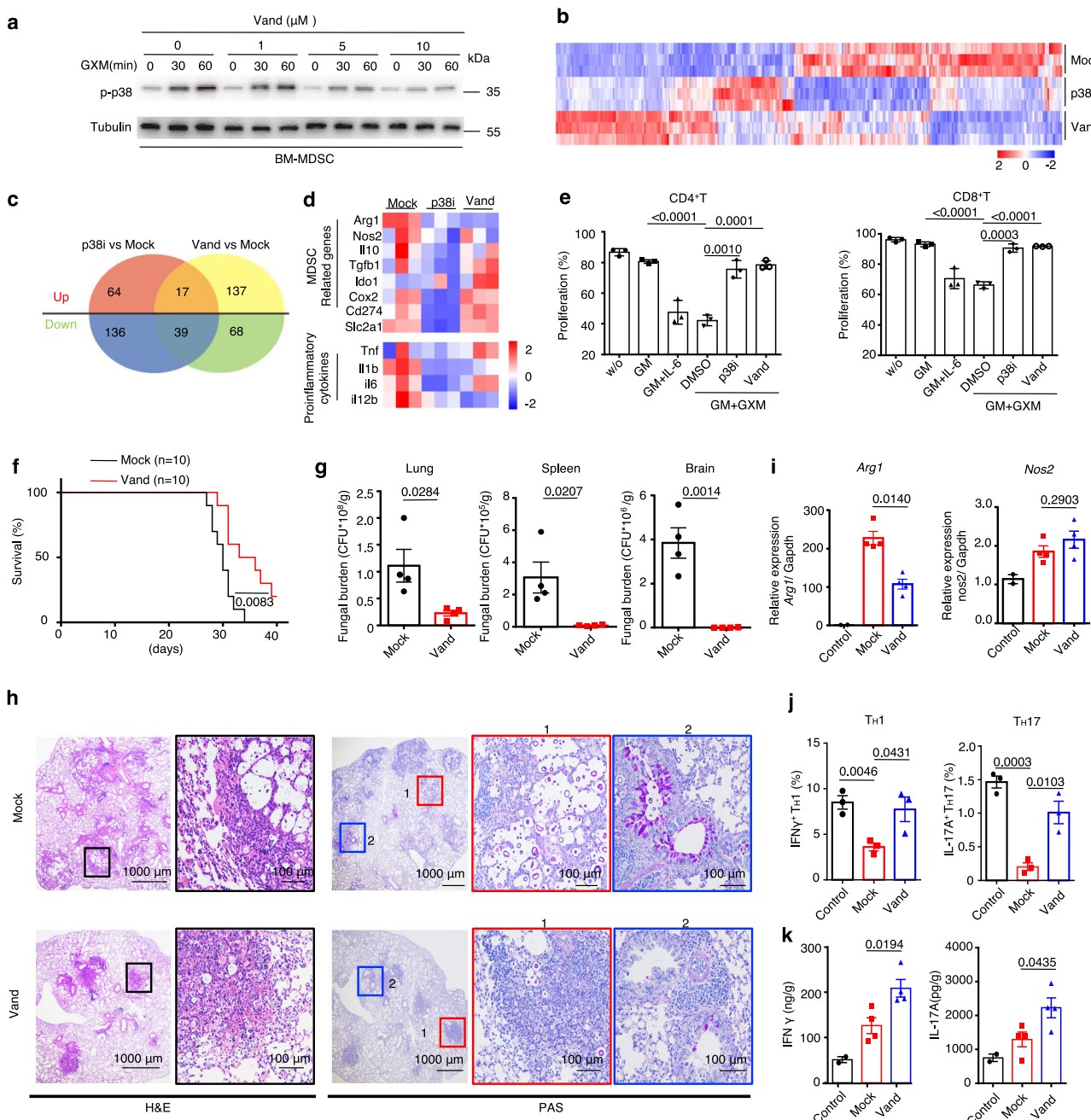

**Fig. 6 Vandetanib inhibits MDSC-derived arginase-1 production to enhance T-cell-based immunotherapy against *C. neoformans* infection. a** Western blot assay of phosphorylated p38 in BM-derived MDSCs, which were generated by GM-CSF (40 ng/ml) + GXM (10 μg/well) for 5 days and then treated with GXM (10 μg/well) combined with vandetanib at the indicated concentration and time, Tubulin (55 kDa) served as a loading control. Representative blots were shown from three independent experiments. **b–d** Heatmap (**b**) and Venn diagram (**c**) of all differentially expressed genes, and heatmap of differentially expressed MDSC-related and pro-inflammatory genes (**d**) in wild-type BM-derived MDSCs, which were generated by GM-CSF (GM, 40 ng/ml) + GXM (10 μg/well) and then treated with GXM (10 μg/well) combined with p38 inhibitor SB202190 (p38i, 10 μM) or vandetanib (Vand, 5 μM) for 3 h. **e** Proliferation frequency of CD4+ and CD8+ T-cells, which were co-cultured with wild-type BM-derived MDSCs with or without treatment of p38 inhibitor SB202190 (p38i, 10 μM) or vandetanib (Vand, 5 μM) for 72 h. **f** Survival assay of *C. neoformans* strain H99-infected mice ($n = 10$, $1 \times 10^3$ CFU/mouse), which were intraperitoneally treated with or without vandetanib (Vand, 50 mg/kg every other day). **g–k** Fungal burden in lung, spleen and brain on Day 21 (**g**), representative histological images with hematoxylin-eosin (H&E) and Periodic Acid-Schiff (PAS) staining (**h**), mRNA levels of *Arg1* and *Nos2* (**i**), the frequency of IFNγ+ TH1 and IL-17A+ TH17 cells (**j**), and the amount of IFN-γ and IL-17A (**k**) in the lungs of *C. neoformans* strain H99-infected mice ($1 \times 10^3$ CFU/mouse) on Day 14, which were intraperitoneally treated with or without vandetanib (Vand, 50 mg/kg every other day). Scar bars = 1000 μm (left) or 100 μm (right). Data were presented as mean ± SEM, $n = 3$ (**e**, **j**), $n = 4$ (**g–i**), and $n = 10$ (**h**) biologically independent samples. Data were analyzed by unpaired two-sided Student's *t*-test in **e**, **g–i**, **j**, **k** and two-sided log-rank (Mantel–Cox) tests in **f**. Source data are provided as a Source Data file.

were immunized with the hCLEC2D-hIgG1-Fc fusion protein. After a final intramuscular boost, the splenocytes were harvested and fused with Y3 myeloma cells. Hybridoma supernatants were screened using ELISA and a synthetic peptide of human CLEC2D (lndkgassarhyterkwics). Specificity of monoclonal antibody were tested using western blot assays, with bone marrow cells as a positive control and Raw264.7 as a negative control. Two specific mAbs clones for mouse CLEC2D (2260CT10.1.3.2 and 2260CT34.3.1.1; both IgG1) were selected for further use.

**RNA-sequencing.** Total RNA was extracted using the mirVana miRNA Isolation Kit (Ambion) following the manufacturer's protocol. RNA integrity was evaluated using the Agilent 2100 Bioanalyzer (Agilent Technologies, Santa Clara, CA, USA). The samples with an RNA integrity number (RIN) ≥7 were subjected to subsequent analysis. The libraries were constructed using the TruSeq Stranded mRNA LT Sample Prep Kit (Illumina, San Diego, CA, USA) according to the manufacturer's instructions. The libraries were sequenced on the Illumina sequencing platform (HiSeqTM 2500 or Illumina HiSeq X Ten), and 125 bp/150 bp paired-end reads were generated at Shanghai OE Biotech Co. Ltd.

**Statistics and reproducibility.** All experiments were performed at least three times. Immunofluorescence staining, immunohistochemical staining, western blot assays, and DNA agarose gel blot representative images are shown. Data were analyzed using GraphPad Prism 8. Unpaired two-tailed Student's $t$-test was used to analyze the differences between two groups. Comparisons among multiple groups were analyzed with one-way ANOVA. Survival curves were compared using the log-rank test. The results are presented as means ± SEM. $P < 0.05$ was considered statistically significant.

**Reporting summary.** Further information on research design is available in the Nature Research Reporting Summary linked to this article.

## Data availability
The sequence data generated in this study have been deposited in the GEO database under the accession code PRJNA799667. All the data generated in this study are provided in the Supplementary Information and Source Data file. Source data are provided with this paper.

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

## Acknowledgements

This work was supported by the National Key Research and Development Program of China (2021YFC2300400 to X.-M.J.), the National Natural Science Foundation of China (31970889 to X.-M.J., 82000002 to Y.-N.L, 81901673 to Z.-W.W., and 8197080313 to L.-P.Z.), Innovation Program of Shanghai Municipal Education Commission (201901070007E00022 to X.-M.J.), the Key fund for basic research of Shanghai Science and Technology Commission (20JC1417700 to X.-M.J.), Shanghai Laboratory Animal Research Fund (21140903300 to X.-M.J.), Cultivation Project of Shanghai Major Infectious Disease Research Base (20DZ2210500 to J.-M.Q.), Shanghai Key Laboratory of Emergency Prevention, Diagnosis and Treatment of Respiratory Infectious Diseases (20DZ2261100 to J.-M.Q.), China Postdoctoral Science Foundation (2019M661549 to Y.-N.L.), the Fundamental Research Funds for the Central Universities to X.-M.J. and Innovative research team of high-level local universities in Shanghai to X.-M.J. and J.-M.Q.

## Author contributions

Y.-N.L., Z.-W.W., F.L., and L.-H.Z. performed the experiments. Y.-N.L. and Z.-W.W. performed the statistical analyses. Y.-S.J., Y.Y., H.-H.M., and L.-P.Z. contributed critical reagents and clinical samples. Y.-N.L., J.-M.Q., and X.-M.J. designed the study. Y.-N.L., Z.-W.W., and X.-M.J. wrote the manuscript.

## Competing interests

The authors declare no competing interests.
