## [Peer Review File · Nature Communications]

Inhibition of myeloid-derived suppressor cell arginase-1 production enhances T-cell-based immunotherapy against *Cryptococcus neoformans* infectionREVIEWER COMMENTS

Reviewer #1 (Remarks to the Author):

The authors have conducted an extremely comprehensive study supporting their hypothesis that C-type lectin receptor-2d (Clec2d) recognizes GXM to potentiate the immunosuppressive activity of neutrophilic MDSCs through initiating p38-mediated production of Arg-1, which inhibits T cell-mediated antifungal responses against *C. neoformans* infection.

I have no major criticisms of the work

General comment:

A grammatical review of the manuscript is needed.

The Figures each show large amounts of data and the pathological sections so small that it is not possible to interpret them with respect to infiltrating cell types or constituent cells in "granulomas" or indeed if they are true granulomas.

Introduction:

The section on the potential clinical relevance of the study as summarised in the introduction needs to be improved.

Refs 2 and 3 (1991 and 2003) need updating and qualification with respect to ART (antiretroviral) therapy

Line 71. A reference for statement that efficacy of antifungal drugs is diminishing due to increased drug resistance and inability to assist the antifungal response is needed. Additional rationale for new therapeutic approaches (not mentioned) is the toxicity and need for IV therapy for the most effective drug (AMB).

Further publications on GXM (and GalX) have appeared since Refs 9 (1987), 10 (1997) and should selectively be referenced. Notably there is debate about whether GXM or GalXM is more important in the immunomodulatory effects of *Cryptococcus* (Decote-Ricardo D et al *Front. Med (Lausanne)*. 2019; 6: 129. PMID: PMC6593061).

In Fig 1 lung recruitment of M-MDSC and PMN-MDSC was assessed whereas blood MDSC were assessed in 15 patients with pulmonary cryptococcosis and 14 with cryptococcal meningitis. It is not stated whether any were immunosuppressed (eg by HIV or other cause) which might influence the results. Although when expressed as a ratio per 10 log 5 PBMC or as a percentage there was a statistically significant increase in PMN-MDSC especially in patients with meningitis, there was wide variation between individuals. These points should receive comment in the discussion.

In general, the different experimental manipulations in the mice significantly impacted fungal burden and pathology, to a greater extent than survival, which was prolonged rather than complete. Taken together however, the authors have shown that that in mice, PMN-MDSCs play a role in respect of amplification of cryptococcal lung disease and that the putative mechanism of action proceeds through Arg-1 as a result of Clec2d recognising components of cryptococcal GXM.

Line 324. Vandetanib is a non-specific tyrosine kinase inhibitor and it is quite a stretch to suggest that it per se is a promising immunotherapeutic for human CM. On the basis of the P38 data in particular, it is reasonable to say that inhibition of this Arg-1 pathway provides a potentially novel immunotherapeutic approach.

Overall this is a well-conducted study which adds a role for PMN-MDSCs to other mechanisms in the pathogenesis of cryptococcal disease, as has been reported in other infectious diseases such as tuberculosis.

Reviewer #2 (Remarks to the Author):

Li and colleagues investigated the effect of *Cryptococcus neoformans* infection and glucuronoxylomannan (GXM; main capsular component) release on neutrophilic myeloid-derived suppressor cells (MDSCs). Release of GXM by the fungus increases recruitment of neutrophilic MDSCs and this is disease enhancing. Clec2d, a C-type lectin receptor, binds to GXM and promotes neutrophilic MDSCs via p38-mediated production of arginase-1 (Arg-1), which impairs T cell responses against the fungus. They also validated the pharmacological use of a p38 inhibitor, SB202190, which reduces the expression of Arg-1 and vandetanib, an inhibitor of tyrosine kinase as stimulants of T cell antifungal responses. The strengths of the study include the huge amount of work performed by the team including the generation of an Clec2d knock out mouse and the use of many techniques (e.g., flow cytometry, RNA seq, histopathology, etc.). However, the premise that *C. neoformans* causes disease in immunocompetent individuals is weak given that cryptococcosis affects mostly T cell deficient individuals such as those with HIV+/AIDS, organ transplantation, etc. It is uncertain how the proposed therapy will work in immunocompromised individuals with deficient cell-mediated immunity. Furthermore, there are many other weaknesses in the study that are described below.

1. The number of deaths per year globally due to cryptococcal meningitis are ~200,000 not ~625,000. This is an old piece of information.
2. In addition to resistance and their inability to induce a strong immune response, antifungal drugs to treat cryptococcosis are toxic, have difficulty penetrating the CNS, and are not accessible in the world regions where are mostly needed. All this is also exacerbated by the fact that individuals with cerebral cryptococcosis are immunosuppressed.
3. Methods are poorly written to reproduce the studies or data interpretation. For example, routes of infection, mouse strains, and many other specifics to understand the studies are missing or insufficiently described.
4. Also, there is a concern with the patient data presented in Fig. 1D. The data points of the graphs look the same and have similar p values even when different parameters are analyzed (e.g., MDSCs/ 10^5 PBMSs vs. % MDSCs in PBMSs). If the same data is analyzed using different parameters, the graphs should be at least slightly different but not the same as presented. This reviewer gives the benefit of the doubt to the authors because unintended mistakes happen particularly when you deal with the amount of data presented in this study. Nevertheless, the methods are so poorly described that it is impossible to understand the analyses performed. Fig. 2D, the third graph to the right is missing the error bars. In fact, the statistics described in the methods are incomplete or not properly explained to understand the details of the studies.
5. Many results are logically presented. However, the discussion does not make justice to the work performed. It is inadequately described or put on perspective with previous or possible future research.
6. The work might be new in cryptococcosis, but similar work was described in *Candida albicans* (PMID: 25771792), therefore, these studies lack novelty because the overall connection between MDSCs and T cells is expected. In addition, it is uncertain the potential application of the therapeutics proposed in the setting of immunosuppression or in cerebral cryptococcosis, the main reason of death in patients. Also, the use of T cell depleted, or deficient mice would have been more appropriate for these studies.

We would like to thank the reviewers for their critical and constructive comments and suggestions. Based on their comments, we have performed all of the requested experiments and revised our manuscript. The following is the point-to-point responses to their comments.

Reviewer #1

The authors have conducted an extremely comprehensive study supporting their hypothesis that C-type lectin receptor-2d (Clec2d) recognizes GXM to potentiate the immunosuppressive activity of neutrophilic MDSCs through initiating p38-mediated production of Arg-1, which inhibits T cell-mediated antifungal responses against *C. neoformans* infection.

I have no major criticisms of the work.

General comment:

1. A grammatical review of the manuscript is needed.

Response: Thank you for your good suggestion. A grammatical review of our revised manuscript has been made by an academic English editor of AJE company.

2. The Figures each show large amounts of data and the pathological sections so small that it is not possible to interpret them with respect to infiltrating cell types or constituent cells in “granulomas” or indeed if they are true granulomas.

Response: We have rearranged Figure 1 and moved the part of original Fig 1A to Supplementary S1B in our revised manuscript. Thus, the pathological sections have been zoomed in (Fig 1H and 1K). Dotted lines indicate the formation sites of granulomas. Black arrows indicate yeast cells. We can observe that PMN-MDSC depletion by Ly-6G antibody led to a low number of yeast cells, which were mostly encapsulated in the granulomatous tissues in the infected lungs.

3. The section on the potential clinical relevance of the study as summarized in the introduction needs to be improved.

Response: According to your suggestion, we have added the potential clinical relevance summarization of our study in Introduction section of our revised manuscript as following: “We showed that release of GXM by *C. neoformans*

increases recruitment of neutrophilic MDSCs thereby aggravating this infectious disease. Clec2d, a CLR, recognized GXM and promoted neutrophilic MDSCs via p38-mediated production of Arg-1, which impaired T cell responses against *C. neoformans* infection. We also validated that pharmacological inhibition of MDSC-derived Arg-1 production significantly enhanced T cell-mediated antifungal responses against *C. neoformans* infection. Together, our data indicate that neutrophilic MDSCs play a role in respect of amplification of cryptococcal lung disease and inhibiting MDSC-derived Arg-1 production is a promising immunotherapeutic strategy for treating this disease.”

4. Refs 2 and 3 (1991 and 2003) need updating and qualification with respect to ART (antiretroviral) therapy.

Response: We have updated references according to your suggestion and added new statement with respect to ART as following: “The effectiveness of antiretroviral therapy (ART) in restoring CD4⁺ T cell numbers in AIDS patients has dramatic reductions in mortality and morbidity of cryptococcal meningitis (Vibhagool et al. Clin Infect Dis, 2003, 36, 1329-1331). However, severe adverse reaction to ART is referred to as immune reconstitution inflammatory syndrome (IRIS), which is characterized by tissue-destructive inflammation and arises as CD4⁺ T cells re-emerge (Barber et al. Nat Rev Microbiol,2012, 10, 150-156).”

5. Line 71. A reference for statement that efficacy of antifungal drugs is diminishing due to increased drug resistance and inability to assist the antifungal response is needed. Additional rationale for new therapeutic approaches (not mentioned) is the toxicity and need for IV therapy for the most effective drug (AMB).

Response: We have added a new reference to the following revised statement: “Recent World Health Organization guidelines recommend 7-day amphotericin B plus flucytosine, then 7-day high dose (1200 mg/day) fluconazole for the treatment of AIDS-associated cryptococcal meningitis (Temfack et al. Curr Neurol Neurosci Rep. 2019, 19, 81). However, the efficacy of antifungal drugs is limited by host toxicity, pathogen resistance, the impenetrability across the

blood-brain barrier and the inability to induce the antifungal response (Iyer et al. Nat Rev Microbiol, 2021, 19, 454-466)”.

6. Further publications on GXM (and GalX) have appeared since Refs 9 (1987), 10 (1997) and should selectively be referenced.

Response: We have updated references with recent publications about GXM as following: “However, Cryptococcus cells prevent recognition and phagocytosis by masking themselves in an exopolysaccharide-based capsule composed primarily of glucuronoxylomannan (GXM) (Roeder et al. Med Mycol. 2004, 42, 485-498). Numerous lines of studies indicate that GXM exhibits potent immunosuppressive properties of inhibiting CD4+ and CD8+T cell-mediated immunity (CMI) (Yauch et al. PLoS Pathog, 2006, 2, e120; Monari et al. FEMS Yeast Res 2006, 6, 537-542).”

7. Notably there is debate about whether GXM or GalXM is more important in the immunomodulatory effects of Cryptococcus (Decote-Ricardo D et al Front. Med (Lausanne). 2019; 6: 129. PMID: PMC6593061).

Response: In this review (Decote-Ricardo D et al Front. Med (Lausanne). 2019; 6: 129.), they summarize that GXM contributes to immune suppression through inducing IL-10 and TGF- β production whereas GalXM can induce the production of pro-inflammatory cytokines TNF- α and IL-6. The common feature is that both GXM and GalXM can induce apoptosis of T cells and macrophages. They conclude that GalXM-mediated cell death could enhance the suppressive effect of GXM during cryptococcosis. We extracted GXM and GalXM from *C. neoformans*, which were confirmed by a commercially available antibody 18B7 (Fig R1A). We further showed that GXM was more potent to induce the tolerogenic activity of MDSCs than that of GalXM, which was evidenced by their different abilities to impair the proliferation of CD4+ and CD8+ T cells (Fig R1B).

Fig R1 (A) ELISA was carried out to test the binding activity of antibody 18B7 to GXM or GalXM. (B) Bone marrow-derived myeloid cells were co-cultured with medium (Ctrl), GM-CSF (GM, 40ng/ml), GM+IL-6 (40ng/ml) GM+GXM (10 μ g/well) or GM+GalXM (10 μ g/well) for 5 days. Quantitation of CD4⁺ or CD8⁺ T cell proliferation, which were co-cultured without (w/o) or with BM-derived MDSCs. Mean \pm SEM from n=3, *P<0.05 and **P<0.01 by Mann-Whitney for pairwise comparison and Kruskal-Wallis for okimulti-group comparison.

8. In Fig 1 lung recruitment of M-MDSC and PMN-MDSC was assessed whereas blood MDSC were assessed in 15 patients with pulmonary cryptococcosis and 14 with cryptococcal meningitis. It is not stated whether any were immunosuppressed (eg by HIV or other cause) which might influence the results. Although when expressed as a ratio per 10 log 5 PBMC or as a percentage there was a statistically significant increase in PMN-MDSC especially in patients with meningitis, there was wide variation between individuals. These points should receive comment in the discussion.

Response: Thank you for your comments. We have added the description about the patients in Discussion section of our revised manuscript as following: “we observed that non-HIV 15 patients with pulmonary cryptococcosis and 14 patients with cryptococcal meningitis (Table S1) significantly accumulated PMN-MDSCs, but not M-MDSCs, in PBMCs compared to healthy controls. Among these patients, 46.7% with pulmonary cryptococcosis and 64.3% with cryptococcal meningitis are immunosuppressed or immunocompromised. There was no significant difference about the accumulation of PMN-MDSCs between immunocompetent and immunosuppressed or immunocompromised patients (Fig R2). However, it remains unclear whether HIV infection affect the accumulation of MDSCs in patients with pulmonary cryptococcosis or cryptococcal meningitis.”

Fig R2 HLA-DR⁺CD11b⁺CD33⁺MDSCs, HLA-DR⁺CD11b⁺CD33⁺CD15⁺CD14⁻ PMN-MDSCs and HLA-DR⁺CD11b⁺CD33⁺CD15⁻CD14⁺ M-MDSCs ratios of PBMCs in Total patients, Immunocompetent or immunocompromised patients in pulmonary cryptococcosis (PC) and cryptococcal meningitis (CM) respectively.

9. In general, the different experimental manipulations in the mice significantly impacted fungal burden and pathology, to a greater extent than survival, which was prolonged rather than complete. Taken together however, the authors have shown that that in mice, PMN-MDSCs play a role in respect of amplification of cryptococcal lung disease and that the putative mechanism of action proceeds through Arg-1 as a result of Clec2d recognising components of cryptococcal GXM.

Response: To evaluate the GXM-induced tolerogenic activity of PMN-MDSCs on the host defense against *C. neoformans* infection, we adoptively transferred PMN-MDSCs sorted from the lungs of mice with or without GXM treatment and monitored their impact on fungal burdens in Cap59-infected mice (Fig R3A). We found that this transfer significantly increased fungal burdens in lungs of Cap59-infected mice (Fig R3B). Moreover, we showed that only adoptive transfer of Clec2d-deficient MDSCs induced by GM-CSF plus GXM, but not GM-CSF plus IL-6, dramatically decreased fungal burdens and Arg-1 expression in the lungs of

Cap59-infected WT mice (Figure 4F-G). Together, we suggest that Clec2d controlled the tolerogenic activity of MDSCs through recognizing GXM to aggravate *C. neoformans* infection.

Fig R3 Sorting M-MDSCs^G or PMN-MDSCs^G from GXM pre-treated mice. Adoptive transferred to wild-type mice, which were intratracheally infected with Cap59 (1×10^5 CFU/mice) for 2hrs later. **(A-B)** The percentage of MDSCs, M-MDSCs or PMN-MDSCs **(A)** or pulmonary fungal burden **(B)** of Cap59-infected mice were shown after adoptive transfer with M- or PMN-MDSCs, respectively. Above representative dot plots from three independent experiments. Mean \pm SEM from n=3, *P<0.05 and **P<0.01 by Mann-Whitney for pairwise comparison.

10. Line 324. Vandetanib is a non-specific tyrosine kinase inhibitor and it is quite a stretch to suggest that it per se is a promising immunotherapeutic for human CM. On the basis of the P38 data in particular, it is reasonable to say that inhibition of this Arg-1 pathway provides a potentially novel immuno-therapeutic approach.

Response: Thank you for your positive comments.

11. Overall this is a well-conducted study which adds a role for PMN-MDSCs to other mechanisms in the pathogenesis of cryptococcal disease, as has been reported in other infectious diseases such as tuberculosis.

Response: Thank you for your positive comments.

Reviewer #2

Li and colleagues investigated the effect of *Cryptococcus neoformans* infection and glucuronoxylomannan (GXM; main capsular component) release on neutrophilic myeloid-derived suppressor cells (MDSCs). Release of GXM by the fungus increases recruitment of neutrophilic MDSCs and this is disease enhancing. Clec2d, a C-type lectin receptor, binds to GXM and promotes neutrophilic MDSCs via p38-mediated production of arginase-1 (Arg-1), which impairs T cell responses against the fungus. They also validated the pharmacological use of a p38 inhibitor, SB202190, which reduces the expression of Arg-1 and vandetanib, an inhibitor of tyrosine kinase as stimulants of T cell antifungal responses. The strengths of the study include the huge amount of work performed by the team including the generation of an Clec2d knock out mouse and the use of many techniques (e.g., flow cytometry, RNA seq, histopathology, etc.). However, the premise that *C. neoformans* causes disease in immunocompetent individuals is weak given that cryptococcosis affects mostly T cell deficient individuals such as those with HIV+/AIDS, organ transplantation, etc. It is uncertain how the proposed therapy will work in immunocompromised individuals with deficient cell-mediated immunity.

Response: Thank you for your comments. As we know, *C. neoformans* prominently causes disease in immunocompromised individuals such as those with AIDS, organ transplantation, etc. Our present study showed that release of GXM by *C. neoformans* increases recruitment of neutrophilic MDSCs, which inhibited the function of T cells, thereby aggravating this infectious disease. Therefore, we propose that our data can partially explain why those immunocompromised individuals are more vulnerable to *C. neoformans* infection in a new angle. We agree that it is uncertain whether our proposed therapy will work in immunocompromised individuals with deficient cell-mediated immunity. However, there are many cases of cryptococcal infections in immunocompetent hosts (Setianingrum et al. *Med Mycol.* 2019, 57(2):133-150; Hu et al. *BMC Pulm Med.* 2021, 21(1):262; Ruan et al. *BMC Infect Dis.* 2017, 17(1):369). On the other

hand, we expect that other international researchers are willing to test the effects of our proposed therapy using a murine model of co-infection with *C. neoformans* and HIV when our manuscript is published.

Furthermore, there are many other weaknesses in the study that are described below.

1. The number of deaths per year globally due to cryptococcal meningitis are ~200,000 not ~625,000. This is an old piece of information.

Response: Thank you for pointing this out. We have revised this statement as following: “An estimated 223,100 cases of cryptococcal meningitis occur globally annually, which lead to about 181,100 deaths.”

2. In addition to resistance and their inability to induce a strong immune response, antifungal drugs to treat cryptococcosis are toxic, have difficulty penetrating the CNS, and are not accessible in the world regions where are mostly needed. All this is also exacerbated by the fact that individuals with cerebral cryptococcosis are immunosuppressed.

Response: Thank you for your good suggestion. We have revised the following introduction: “However, the efficacy of antifungal drugs is limited by host toxicity, pathogen resistance, the impenetrability across the blood-brain barrier and the inability to induce the antifungal response (Iyer et al. Nat Rev Microbiol, 2021, 19, 454-466)”.

3. Methods are poorly written to reproduce the studies or data interpretation. For example, routes of infection, mouse strains, and many other specifics to understand the studies are missing or insufficiently described.

Response: Thank you for your critical comments. We have added a more detail description of methods in our revised manuscript, which was highlighted with red color in the section of Methods.

4. Also, there is a concern with the patient data presented in Fig. 1D. The data

points of the graphs look the same and have similar p values even when different parameters are analyzed (e.g., MDSCs/10⁵ PBMSs vs. % MDSCs in PBMSs). If the same data is analyzed using different parameters, the graphs should be at least slightly different but not the same as presented. This reviewer gives the benefit of the doubt to the authors because unintended mistakes happen particularly when you deal with the amount of data presented in this study.

Response: Thank you for pointing it out. Actually, we made a mistake about p value of analyzing MDSCs number in patient data. We have corrected them and moved original part of Fig. 1D to Supplementary Fig 1E.

5. Nevertheless, the methods are so poorly described that it is impossible to understand the analyses performed. Fig. 2D, the third graph to the right is missing the error bars. In fact, the statistics described in the methods are incomplete or not properly explained to understand the details of the studies.

Response: We have added the missing error bars in Fig 2B, 2D and 5M. We also added the description of statistics in the section of Methods as following: “data was expressed as mean ± SEM, Differences between two groups were compared using Mann–Whitney (GraphPad Prism, La Jolla, CA, USA), and grouped comparison was evaluated by non-parametric ANOVA and subsequent Kruskal–Wallis (Kruskal–Wallis). Statistical significance was set on the basis of P value. n.s., *P < 0.05, **P < 0.01, *P < 0.001.”**

6. Many results are logically presented. However, the discussion does not make justice to the work performed. It is inadequately described or put on perspective with previous or possible future research.

Response: Thank you for your good suggestion, we have modified our discussion to add more detail description and perspectives in our revised manuscript, which were highlighted with red color.

7. The work might be new in cryptococcosis, but similar work was described in *Candida albicans* (PMID: 25771792), therefore, these studies lack novelty because the overall connection between MDSCs and T cells is expected.

Response: In this previous study, they show that beta-glucans exposed on *Candida albicans* can induce neutrophilic MDSCs through Dectin-1 and its downstream adaptor protein CARD9 and neutrophilic MDSCs were protective from systemic *C. albicans* infection. In our present study, we showed that GXM from *C. neoformans* induced the tolerogenic activity of neutrophilic MDSCs to aggravate their infections. More importantly, we identified a new receptor Clec2d of MDSCs to recognize GXM for controlling their tolerogenic activity. Based on the Clec2d/p38/Arg-1 data, we further suggest that inhibition of this Arg-1 pathway provides a potentially novel immuno-therapeutic approach against cryptococcal disease.

8. In addition, it is uncertain the potential application of the therapeutics proposed in the setting of immunosuppression or in cerebral cryptococcosis, the main reason of death in patients. Also, the use of T cell depleted, or deficient mice would have been more appropriate for these studies.

Response: Thank you for your critical comments. According to your suggestions, we used anti-CD4 antibody to deplete CD4⁺T cells in murine model of *C. neoformans* infection (Fig. R4A). Although we observed that the depletion of CD4⁺T cells significantly increased the percentage of CD8⁺T cells in infected lungs, this depletion significantly increased fungal burden in lungs of infected mice (Fig. R4B). Moreover, CD4⁺T cell depletion completely impaired the effects of vandetanib treatment on reducing fungal burden of infected lungs (Fig. R4B). Together, these data confirmed that therapeutic efficacy with vandetanib against *C. neoformans* infection was dependent on CD4⁺T cells. We have also added your suggested comments to the Discussion section in our revised manuscript.

Fig R4 (A) lung CD4⁺, CD8⁺ T cells and (B) lung fungal burdens in mice, which were divided randomly into IgG and anti-CD4 (αCD4) groups intratracheally receiving infection with a single exposure of *C. neoformans* strain H99 (1×10³ CFU/mouse), with or without Vandetanib treatments for three weeks.

REVIEWER COMMENTS

Reviewer #1 (Remarks to the Author):

The authors have satisfactorily answered most of my queries and have amended the manuscript accordingly.

The following still need to be addressed.

Original issue 4.

Refs 2 and 3 (1991 and 2003) need updating and qualification with respect to ART (antiretroviral) therapy.

Response

We have updated references according to your suggestion and added new statement with respect to ART as following: The effectiveness of antiretroviral therapy (ART) in restoring CD4+ T cell numbers in AIDS patients has dramatic reductions in mortality and morbidity of cryptococcal meningitis⁴ However, severe adverse reaction to ART is referred to as immune reconstitution inflammatory syndrome, which is characterized by tissue-destructive inflammation and arises as CD4+ T cells re-emerge⁵. Therefore, there is an urgent need to develop new immunotherapeutic strategies for restoring the function of CD4+ T cells to combat cryptococcosis.

Comment

The logic of the additional sentences is confusing to me. Active cryptococcal meningitis has a significant risk of causing IRIS in patients with AIDS and low CD4 counts if ART therapy is commenced too early (before the fungal infection has been controlled). Although the authors assert that "there is an urgent need to develop new immunotherapeutic strategies for restoring the function of CD4+ T cells to combat cryptococcosis" the manuscript is about "enhanced T cell-mediated antifungal responses against *C. neoformans* infection by an effect on existing CD4 cells. The problem in HIV-infected patients who make up the majority of those with CM globally is that their CD4 cells are low to very low and may not therefore be able to respond to the immunotherapeutics under consideration. This paragraph needs rewording.

Having said this, notably, especially, in resource rich countries, other causes of immunosuppression and additional risk factors, not necessarily associated with low CD4 counts/function are important - and responses to the immune-therapeutics may be advantageous for CM. There is a hint that responses of the immunosuppressed and non-immunosuppressed human subjects are similar in the study (although the sample sizes are very small).

P5 line 130

Start results of the human studies on a new line. Suggest re-word to clarify that all 29 patients were HIV-negative and give the total numbers of immunosuppressed and non-immunosuppressed with PC and CM in the text. The differences between immunosuppressed or would be fine in fig (as this information is highly relevant to clinical context) but low sample size needs a comment.

Discussion: lines 432-437 need to qualify the statement by specifying that there were small numbers of immunosuppressed and non-immunosuppressed human subjects. I note that the lack of information in the context of HIV-associated CD4 deficiency has been commented on (and indeed it should be stated that this may prevent the immunotherapy strategy working).

The real strength of the study is the biology rather than likely therapeutic applications. Grammatical edits will be needed if accepted.

Reviewer #2 (Remarks to the Author):

My comments were diligently addressed by the authors.

Reviewer #3 (Remarks to the Author):

I did not review this paper before, so I carefully examined both manuscript and authors response. I believe, although overall concept of the study is not novel, the specific information regarding Glec2d regulation of MDSC function is novel and interesting. Experiments are well performed. Although mechanism linking p38 and Glec2d as well as p38 and arginase expression remained unclear, this can be developed in subsequent studies.

The main problem with the study is that authors did not provide direct evidence that cells they observed in their model (Figure 1) are indeed MDSC. Infection would cause mobilization of classical neutrophils so this is rather important question. Surface markers used by the authors cannot discriminate between PMN and PMN-MDSC as was demonstrated in the papers authors referred to. So functional evidence of suppression would be important. At minimum, detailed gene expression profile confirming that pathological nature of these cells. Authors used suppression assay later in the paper where they assess the effect of GXM. However, it was done with in vitro generated rather artificial cells (GM-CSF+IL-6), which underscore need for some ex vivo experiment.

The other issue that need correction is the fact that in Figure 4 authors observed only partial abrogation of suppressive activity in KO cells. This fact was not explained. It seems that presented mechanism only partially responsible for the observed phenomenon.

We would like to thank the reviewers for their critical and constructive comments and suggestions. Based on their comments, we have revised our manuscript. The following is the point-to-point responses to their comments.

Reviewer #1 (Remarks to the Author):

The authors have satisfactorily answered most of my queries and have amended the manuscript accordingly.

The following still need to be addressed.

Original issue 4.

Refs 2 and 3 (1991 and 2003) need updating and qualification with respect to ART (antiretroviral) therapy.

Response

We have updated references according to your suggestion and added new statement with respect to ART as following: The effectiveness of antiretroviral therapy (ART) in restoring CD4+ T cell numbers in AIDS patients has dramatic reductions in mortality and morbidity of cryptococcal meningitis. However, severe adverse reaction to ART is referred to as immune reconstitution inflammatory syndrome, which is characterized by tissue-destructive inflammation and arises as CD4+ T cells re-emerge⁵. Therefore, there is an urgent need to develop new immunotherapeutic strategies for restoring the function of CD4+ T cells to combat cryptococcosis.

Comment

The logic of the additional sentences is confusing to me. Active cryptococcal meningitis has a significant risk of causing IRIS in patients with AIDS and low CD4 counts if ART therapy is commenced too early (before the fungal infection has been controlled). Although the authors assert that “there is an urgent need to develop new immunotherapeutic strategies for restoring the function of CD4+ T cells to combat cryptococcosis” the manuscript is about “enhanced T cell-mediated antifungal responses against *C. neoformans* infection by an effect on existing CD4 cells. The problem in HIV-infected patients who make up the majority of those with CM globally is that their CD4 cells are low to very low and may not therefore be able to respond to the immunotherapeutic under consideration. This paragraph needs rewording.

Response: Thank you for your good suggestions. We modified this paragraph according to your suggestions as following: “Clinical and experimental data have

established that CD4⁺T cell-mediated immunity (CMI) is essential for the control of cryptococcal infection since this disease occurs mainly in those with impaired CMI including HIV-infected individuals, with an incidence of 30% and a mortality of 30 to 60%. Notably, antiretroviral therapy (ART), which is applied to restore CD4⁺T cell numbers in AIDS patients, can dramatically reduce the mortality and morbidity of AIDS-associated cryptococcal meningitis. However, a hallmark of infection with *C. neoformans* is the depression of the immune system characterized by poor CMI and pro-inflammatory responses.”

Having said this, notably, especially, in resource rich countries, other causes of immunosuppression and additional risk factors, not necessarily associated with low CD4 counts/function are important - and responses to the immune-therapeutics may be advantageous for CM. There is a hint that responses of the immunosuppressed and non-immunosuppressed human subjects are similar in the study (although the sample sizes are very small).

Response: Thank you for your comments. We agree that it is uncertain whether our proposed therapy will work in immunocompromised individuals with deficient cell-mediated immunity. Our present study showed that release of GXM by *C. neoformans* increases recruitment of neutrophilic MDSCs, which inhibited the function of T cells, thereby aggravating this infectious disease. Our data might partially explain the reason that the immunocompromised individuals are more vulnerable to *C. neoformans* infection in a new angle. On the other hand, there are many cases of cryptococcal infections in immunocompetent hosts (Setianingrum et al. Med Mycol. 2019, 57(2):133-150; Hu et al. BMC Pulm Med. 2021, 21(1):262; Ruan et al. BMC Infect Dis. 2017, 17(1):369).

P5 line 130

Start results of the human studies on a new line. Suggest re-word to clarify that all 29 patients were HIV-negative and give the total numbers of immunosuppressed and non-immunosuppressed with PC and CM in the text. The differences between immunosuppressed or would be fine in fig (as this information is highly relevant to clinical context) but low sample size needs a comment.

Response: We have added the information about the immunosuppressed patients with PC and CM in the section of Results as following: Among these patients, 7 with pulmonary cryptococcosis and 9 with cryptococcal meningitis are immunosuppressed or immunocompromised. However, there was no significant difference about the accumulation of PMN-MDSCs between immunocompetent and immunosuppressed or immunocompromised patients (Figure S1G).

Discussion: lines 432-437 need to qualify the statement by specifying that there were small numbers of immunosuppressed and non-immunosuppressed human subjects. I note that the lack of information in the context of HIV-associated CD4 deficiency has been commented on (and indeed it should be stated that this may prevent the immunotherapy strategy working).

Response: We add the comments in the section of Discussion according to your suggestions: “However, it needs further studies to explore whether enhancing the function of T cells through inhibiting MDSC-derived Arg-1 production could reduce the mortality and morbidity of AIDS-associated cryptococcal meningitis since these patients have very few CD4⁺T cells and may not therefore be able to respond to our proposed T cell-mediated immunotherapy.”

Reviewer #2 (Remarks to the Author):

My comments were diligently addressed by the authors.

Response: Thank you for your positive comments.

Reviewer #3 (Remarks to the Author):

I did not review this paper before, so I carefully examined both manuscript and authors response. I believe, although overall concept of the study is not novel, the specific information regarding Glec2d regulation of MDSC function is novel and interesting. Experiments are well performed. Although mechanism linking p38 and Glec2d as well as p38 and arginase expression remained unclear, this can be developed in subsequent studies.

The main problem with the study is that authors did not provide direct evidence that cells they observed in their model (Figure 1) are indeed MDSC. Infection would cause mobilization of classical neutrophils so this is rather important question. Surface markers used by the authors cannot discriminate between PMN and PMN-MDSC as was demonstrated in the papers authors referred to. So functional evidence of suppression would be important. At minimum, detailed gene expression profile confirming that pathological nature of these cells. Authors used suppression assay later in the paper where they assess the effect of GXM. However, it was done with in vitro generated rather artificial cells (GM-CSF+IL-6), which underscore need for some ex vivo experiment.

Response: Thank you for your constructive suggestions. Actually, we have provided the functional evidence of MDSC suppression, which was included in

Figure 2. To highlight these data, we re-arranged the data in Figure 2 as following: “We further evaluated the GXM-induced tolerogenic activities of MDSCs both *ex vivo* and *in vivo* (Figure 2E). We found that MDSCs sorted from the lungs of GXM-treated mice strongly suppressed CD4⁺ and CD8⁺T cell proliferation in a dose-dependent manner (Figure 2F and S2E). Moreover, we found that the adoptive transfer of MDSCs from the lungs of GXM-treated mice significantly increased the percentage of MDSCs, especially PMN-MDSCs, and fungal burdens in the lungs of Cap59-infected mice (Figure 2G-H and S2F). Notably, the adoptive transfer of MDSCs sorted from the lungs of GXM-treated mice significantly increased the expression of Arg-1, but not iNOS, in the lungs of Cap59-infected mice (Figure 2I). These data indicated that GXM induced the tolerogenic activity of MDSCs to inhibit T cell-mediated antifungal responses and aggravate *C. neoformans* infection.”

The other issue that need correction is the fact that in Figure 4 authors observed only partial abrogation of suppressive activity in KO cells. This fact was not explained. It seems that presented mechanism only partially responsible for the observed phenomenon.

Response: According to your good suggestions, we have added the comments in the section of Discussion of our revised manuscript as following: “More importantly, we showed that adoptive transfer of Clec2d-deficient MDSCs induced by GM-CSF plus GXM into wild-type mice significantly decreased their pulmonary fungi burden and Arg-1 expression after infection with *C. neoformans* hypocapsular strain Cap59. In contrast, adoptive transfer of wild-type MDSCs induced by GM-CSF plus GXM into Clec2d-deficient mice significantly increased their pulmonary fungi burden and Arg-1 expression to suppress the function of IFN- γ -producing T_H1 cells and IL-17A-producing T_H17 cells after infection with *C. neoformans* encapsulated strain H99. These data suggested that Clec2d partially controlled the tolerogenic activity of MDSCs through recognizing GXM to aggravate *C. neoformans* infection. Further studies are needed to determine whether other receptors are involved in the recognition of GXM from *C. neoformans* to affect the suppressive activities of MDSCs.”

REVIEWER COMMENTS

Reviewer #1 (Remarks to the Author):

The authors have now satisfactorily addressed my concerns

Reviewer #3 (Remarks to the Author):

Authors addressed all my concerns satisfactory.

POINT BY POINT REPLY TO THE REVIEWERS' COMMENTS (reproduced verbatim)

Reviewer #1 (Remarks to the Author):

The authors have now satisfactorily addressed my concerns

Reviewer #3 (Remarks to the Author):

Authors addressed all my concerns satisfactory.

Response: We are glad to know that Reviewer #1 and #3 had no further concerns and were satisfied with our efforts to address all their questions. We would like to take this occasion to thank them once again for all constructive suggestions, which helped us to improve our manuscript.